# Harmonics Signal Feature Extraction Techniques: A Review

**Minh Ly Duc** [1,2,*] , **Petr Bilik** [2] and **Radek Martinek** [2]

1    Faculty of Commerce, Van Lang University, 69/68 Dang Thuy Tram, Ward 13, BinhThanh District,
     Ho Chi Minh City 70000, Vietnam
2    Department of Cybernetics and Biomedical Engineering, VSB–Technical University of Ostrava, 17. Listopadu
     15, 708 33 Ostrava, Czech Republic; petr.bilik@vsb.cz (P.B.); radek.martinek@vsb.cz (R.M.)
*    Correspondence: minh.ld@vlu.edu.vn

**Abstract:** Harmonic estimation is essential for mitigating or suppressing harmonic distortions in power systems. The most important idea is that spectrum analysis, waveform estimation, harmonic source classification, source location, the determination of harmonic source contributions, data clustering, and filter-based harmonic elimination capacity are also considered. The feature extraction method is a fundamental component of the optimization that improves the effectiveness of the Harmonic Mitigation method. In this study, techniques to extract fundamental frequencies and harmonics in the frequency domain, the time domain, and the spatial domain include 67 literature reviews and an overall assessment. The combinations of signal processing with artificial intelligence (AI) techniques are also reviewed and evaluated in this study. The benefit of the feature extraction methods is that the analysis extracts the powerful basic information of the feedback signals from the sensors with the most redundancy, ensuring the highest efficiency for the next sampling process of algorithms. This study provides an overview of the fundamental frequency and harmonic extraction methods of recent years, an analysis, and a presentation of their advantages and limitations.

**Keywords:** harmonic; frequency domain; time domain; fundamental frequency

## 1. Introduction

Distributed energy resources have increased the penetration rate of renewable energy sources but have also led to intermittency and poor power quality [1]. To address this, a microgrid combining partially distributed energy resources with a utility grid [2] has been proposed. Harmonic distortion has been proposed to increase the additional losses of electrical equipment, overheating it and reducing equipment efficiency and utilization. The harmonic problem of the microgrid has become a major issue with two main sources: electronic power devices and nonlinear loads [3]. Electronic power devices, such as inverters, rectifiers, and static compensators, which generate high-frequency harmonics that can be suppressed by LC or LCL filters [4], are widely used. Nonlinear loads are the main reason for generating output voltage drop, which leads to the distortion of the inverter output voltage waveform. To reduce harmonics and improve system efficiency, anthropological compensation strategies have been studied [3].

Harmonics affect power quality and increase system losses by up to 27%. Power quality issues are manifested in voltage, current, or frequency deviations, resulting in the failure or malfunction of equipment [5]. Common power issues are temporary or steady-state voltage or frequency variations such as impulsive or oscillatory transients and voltage sags. Voltage sags and dips are caused by short circuit faults and motor starting [6]. Harmonics derate transformers and affect high-frequency controllers, while transients and voltage sag influence protection and control equipment. Alternating current drives ride through interruptions, but induction motor starters and DC drive contactors require backup RC circuits [7].

Extracting the fundamental component of harmonics using traditional and modern techniques is a research trend. It determines the exact harmonic type and is an input to the control algorithms to select the appropriate compensating current for the lost current in the source [3]. The shunt adaptive power filter (SAPF) is a suitable choice for the trend of using modern optimization techniques in the selection of compensating currents, providing high efficiency for compensating the current loss caused by harmonics [8]. Reactive power compensation is the administration of reactive energy to improve the performance of the AC system. It is seen in two ways: load and voltage support. The aim is to achieve an improved power factor and real power balance, while voltage support is necessary to reduce voltage fluctuations at a given terminal [9]. In both cases, the reactive power that flows through the microgrid must be effectively controlled and compensated. Active harmonic filters work on the principle of measuring the magnitude and frequency of the currents (from 1st order to 50th order) of the load [3,8]. The processor will analyze the data and send a signal to control IGBT opening and closing and to bring harmonic currents from the 2nd order to 50th order with the same magnitude and opposite direction as the system harmonic current to eliminate all harmonic currents after the position of connection to the electrical system of the active harmonic filter (Figure 1). The processor performs analysis and extraction algorithms for harmonics more accurately and faster, and the harmonic removal efficiency increases accordingly.

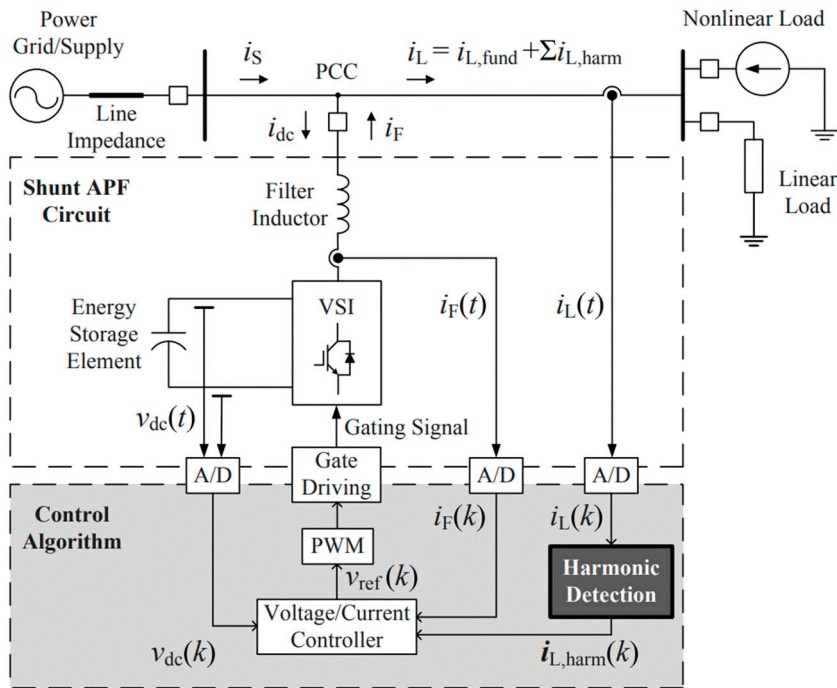

**Figure 1.** Shunt adaptive power filter (SAPF) in three-phase power supply.

Synthetic evaluation studies on the frequency domain and time domain harmonic component extraction methods have been carried out (Table 1). However, there are still many methods of extracting harmonic components in the signal that have not been evaluated by researchers and graduate students. This study conducts a literature review of studies on the frequency domain (the Adaptive Harmonic Wavelet Transform (AHWT) method and Sliding Discrete Wavelet Transform (SDWT) method), time domain (the Empirical Mode Decomposition (EMD) method, Sliding Window EMD (LWEMD) method, Adaptive Harmonic Decomposition (AHD) method, and Adaptive Model-Based Scheme With Short Sliding Analysis Window (AMS) method), and space domain (Head-Related Transfer Functions (HRTF) method) harmonic extraction to provide the most comprehensive overview and a document for future researchers.

**Table 1.** A brief overview of previous literature review documents on methods of extracting harmonics in signals.

| Domain | Methods of Extracting | Feature | Ref | Years |
|---|---|---|---|---|
| Time domain | Selective Harmonic Elimination Pulse-Width Modulation (SHEPWM) | Power signal | [10] | 2017 |
| | Statistical Time-Domain Features method, Upper and Lower Bound of Histogram method, Autoregressive (AR) Coefficients method, Hjorts' Parameters method, Singular Value Decomposition (SVD) method, Piecewise Aggregate Approximation (PAA), and Adaptive Piecewise Constant Approximation (APCA) method | Vibration signal | [11] | 2017 |
| | Mathematical Morphology (MM) Operators method | Electroencephalogram (EEG) signal | [11] | 2017 |
| | SRF algorithm, pq Theory algorithm | Power signal | [12] | 2017 |
| | Fitting a Sum of Exponentials method, Fitting a Straight Line to the Later Stage method | Pulsed Eddy Current (PEC) signal | [13] | 2019 |
| | ADALINE Technique, Self-Tuning Filter (STF) method | Power signal | [14] | 2019 |
| | Convolutional Neural Network (CNN) | Raw signal | [15] | 2020 |
| | Particle Swarm Optimization (PSO), | Power signal | [16] | 2020 |
| | Zero-Crossing Rate (ZCR) method, Short Time Energy (STE) method, Auto-Correlation-Based Features method, Rhythm-Based method | Audio signal | [17] | 2020 |
| Frequency Domains | Fast Fourier Transform (FFT) Method, Eigenvector methods (EM), Wavelet Transform (WT), and Auto-Regressive method (ARM) | Electroencephalogram (EEG) signal | [18] | 2014 |
| | Wavelet Transforms (WT) method, Independent Component Analysis (ICA) method, Principal Component Analysis (PCA) method | Electroencephalogram (EEG) signal | [19] | 2015 |
| | Statistical Frequency-Domain Features, Spectral Skewness, Spectral Kurtosis, Spectral Entropy and Shannon Entropy Feature methods, Short-Time Fourier Transform (STFT) method, Wavelet Transform and Wavelet Decomposition methods, Wigner–Ville Distribution (WVD) method | Vibration signal | [11] | 2017 |
| | Learning Techniques (Genetic Algorithm (GA)) and Artificial Neural Network (ANN), Sliding Window Fourier Analysis (SWFA) | Power signal | [12] | 2017 |
| | Short-Time Fourier Transform (STFT) method, Continuous Wavelet Transform (CWT) method, Hilbert–Huang Transform (HHT) method | Electroencephalogram (EEG) signal | [20] | 2017 |
| | Linear Predictive Coding (LPC) Coefficients method, Code Excited Linear Prediction (CELP) method, Linear Spectral Frequency method | Audio signal | [17] | 2020 |
| | Permutation Entropy (PE) method, Dispersion Entropy (DE) method, Empirical Wavelet Transform (EWT) method, Reverse Dispersion Entropy (RDE) method | Raw signal | [21] | 2020 |
| | Fourier Transform (FT) method, Fast Fourier Transform (FFT) method, S-Transform method, Wavelet-Transform (WT) method | Power signal | [22] | 2021 |
| Space domain | Fractal Dimension method, Correlation Dimension method, Approximate Entropy method, Largest Lyapunov Exponent method, Kolmogorov–Smirnov Test method | Vibration signal | [11] | 2017 |

The main characteristics of the power signal include the phase angle ($\theta$) and magnitude amplitude. The phase origin is calculated from the fundamental component values of the voltage signal ($V_{abc}$ for three-phase sources). In practice, the power supply supplies many loads, including linear and non-linear loads. Non-linear loads include power frequency converters, power supply switches, and LED lighting systems [22]. They are the main source of harmonics in the power supply. Harmonics generated in the power supply cause phenomena such as transformer explosion, heating on the surface of electrical equipment, and the reduced operating efficiency of electrical equipment using power due to source quality supplied with poor quality power [23]. Therefore, determining the lost current correctly and accurately and selecting the correct, sufficient amount of current to compensate for the number of current losses in the power supply generated by the harmonics is a promising future study area for researchers. Analyzing, extracting, and detecting harmonic characteristics play an important role in harmonic mitigation [24]. There has been much research on harmonic mitigation methods in the last few decades. However, the effectiveness of these studies is still limited. Finding a method to implement the harmonic mitigation that brings the most optimal effect is still an open issue for future researchers.

In an application using a shunt adaptive power filter (SAPF) in a three-phase power supply (Figure 1), the lost source current ($i_L$) is compensated by the current extracted from the SAPF ($i_F$); the supply current ($i_S$) is affected by the harmonics arising from the non-linear load. The harmonic processing block provides an algorithm for handling load current ($i_L$) and extracting fundamental frequency and harmonic current ($i_{L,harm}$). The voltage/current controller then generates a PWM pulse from the reference voltage signal $\left( V_{ref} \right)$ fed to the SAPF filter. In some cases, harmonic voltage needs to be detected by methods such as a series adaptive power filter or hybrid adaptive power filter and distributed generation to implement power quality improvement [25,26].

Two methods of detecting harmonics in power supply by extracting harmonics are considered:

- Detecting the overall harmonic, i.e., performing the removal of the fundamental frequency component of the load current (IL), which extracts only the harmonics in the form of a signal [27].
- Selective harmonic detection is the practice of isolating harmonics into a set of signals and extracting them at the output [28,29].

The evaluation of the overall harmonic detection method, the selective harmonic detection method, has many advantages:

- Controlling investment costs for harmonic compensation and improving power quality in a reasonable way [5,30].
- The compensation system has delay time, trigger time, and delay time corresponding to different delay angles of different harmonic types. Selective harmonic compensation allows individual signals to be corrected since each harmonic parameter is adjusted from a single offset angle relative to the hysteresis angle [28].
- The shunt adaptive power filter can be installed in combination with a passive filter that performs the system's hybrid compensation function. The SAPF ensures a low-order harmonic function and the low-pass filter (LPF) performs a high-order harmonic compensation function [31]. However, the LPF is large and does not respond to low-order harmonics. The SAPF has the limitation of not responding to high-order harmonics (high frequency) and the SAPF has a high switching frequency [32]. This increases the electromagnetic interference and insulation stress. In the case of using a hybrid compensation method, it is necessary to select the corresponding reference signal for the SAPF [33].

The fast, accurate harmonic extraction method helps the filter to identify and provide suitable compensation for the lost current in the power supply quickly because harmonics change frequently in the power supply [34]. From different devices, the issues of different grid phases and, correspondingly, different harmonic detection arise but they are both

closely related to real-time harmonic extraction and detection in the power supply. The mesh phase detection method performs the separation of the positive sequence into the basic signal component from the noisy signal [35]. The selective harmonic detection method performs the function of extracting individual harmonics from the current or voltage signal of the disturbed signal [36]. The main objectives of this study include:

- A review of studies related to harmonic feature extraction in the last few decades.
- The frequency and time domain harmonic feature extraction methods and hybrid methods in harmonic feature extraction operation are systematically reviewed.
- The formulas and mathematical models used for the extraction of harmonics in the frequency and time domains are studied and evaluated in detail in this study.
- The overall evaluation and comparison of the processing time efficiency of each harmonic extraction method in the frequency domain and the time domain.
- The identification of the limitations of the harmonic extraction methods in the frequency domain and in the time domain. At the same time, it raises open issues for future research.

This research paper is structured as follows: Section 2 presents the effects of harmonics. Section 3 shows the content of harmonic signal analysis. Section 4 details the harmonic feature extraction technique. Section 5 demonstrates the comparison high light on the harmonic feature extraction technique and the future research topic and Section 6 describes the content of the conclusion.

## 2. Effects of Harmonics

Harmonics are a form of interference that directly affect power quality and have a very bad effect on the equipment and machinery used in a factory. Harmonics cause cables to overheat, damaging insulation [37]. Harmonics reduce motor life, cause motor overheating, and induce a loud operating noise [38]. Harmonics give rise to CB overload, overheating, and transformer explosion (while the amount of electricity used is still less than rated). Harmonics cause circuit breakers, aptomats, and fuses to be affected for unknown reasons [39]. Harmonics cause serious harm to the capacitor by damaging the dielectric, bulging the capacitor, reducing the life of the capacitor, and even causing an abnormal capacitor explosion [40]. Harmonic interference affects telecommunications equipment and automation systems. Harmonics cause measuring equipment to operate incorrectly, and they cause energy waste too (Figure 2). Harmonics occur when the diode rectifier has no passing current and the current goes directly to the inverters while the AC voltage at the input is less than the DC voltage at the capacitor. The sinusoidal shape of the source current is completely distorted when a case of harmonics arises for only a few seconds. The harmonics generated in the power supply cause heating of the conductors, the insulation is broken, the performance of the electrical equipment is reduced, and the life of the electrical equipment is reduced over time. The motor of electrical equipment operates with noise and it is easy to generate heat harmonics that are not well controlled and can damage the dielectrics in the capacitors, shorten the life of the capacitors, and potentially blow up the capacitors. Harmonic currents in rotating machines cause heating effects such as eddy current losses proportional to the square of the frequency [3,8]. Harmonic cycles can cause additional losses by inducing higher frequency currents and negative torques in machine rotors. Harmonic currents can lead to the overloading of power factor correction capacitors and the derating of cables. Harmonic components add phantom power to the total power consumption of a transformer, causing it to overload, heat up, and burn. They heat up and burn conductors, causing serious losses in the electrical system. In a three-phase system, the neutral conductor is heated or burned to create a stable system. The N-G (neutral-earth) voltage is too large. The breaker jumps for unknown reasons. This causes the failure of the PF reactive power compensation capacitor. Noise in communication systems can lead to the overload of capacitors and transformers due to the weakening of harmonic currents, resulting in the formation of an LC circuit. As for systems using backup generators or used on ships and drilling rigs, when running, due to the generator's

inductance characteristics that are higher than conventional transformers, harmonics will be amplified more seriously, from 3 to 4 times, and the seriousness for the system equipment is greater and can even cause a generator fire, which is very dangerous and costly. Harmonics also cause losses on the coil and steel core of the motor to increase, distort the torque form, reduce machine efficiency, and cause noise to affect the error of measuring devices, leading to erroneous measurement results. More dangerously, the higher-order harmonic waves can also generate motor shaft torque or cause mechanical resonance oscillations that damage mechanical components in the engine, causing the flickering of electrical equipment and lighting, affecting people, and causing electromagnetic waves to propagate in space, affecting transceivers.

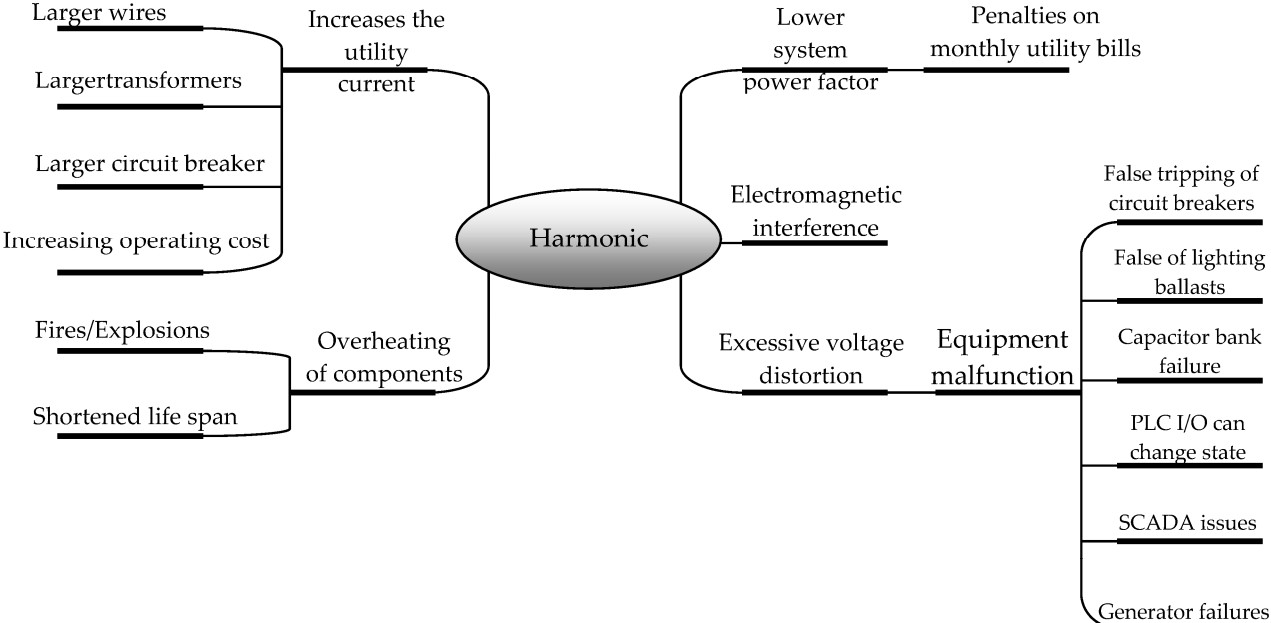

**Figure 2.** Effects of harmonics.

The load in the distribution power supply generates many types of harmonics, affecting the quality of use and the performance of power-using equipment, reducing its efficiency. Much electrical equipment damage, such as fire or explosion, is also caused by harmonic sources [37]. The higher frequency harmonic current causes electrons to flow to the outside of the conductor, which reduces the current-carrying capacity, resulting in a decrease in power rating causing heat gain and damage to the insulation. Harmonic distortion has a direct effect on the power factor [38,39]. Many harmonics have a low power factor value. The heat losses generated by the harmonics shifting to use and pay for the reactive power and harmonic currents can cause the capacitor to fail [33,39].

Transformer aging or heating on the surface of the transformer body is mainly caused by harmonics in the power supply [38]. The transformer structure is formed by winding several coils placed close to each other and separated by insulation; when the power flows through the windings with harmonics generated in them, overvoltage results [34]. Load occurs in the transformer, generating heat in the transformer body, reducing the operating efficiency of the transformer, and reducing the insulation strength of the windings in the transformer. Eddy currents due to stray flux losses cause overheating. A temperature increase of 7–10 degrees can reduce the life of an insulating material by half.

The protection of electrical equipment is provided in the electrolytic power supply devices that perform overload protection, short circuit protection, or protection from overheating generated in electrical circuits, eliminating all the effects potentially affecting the performance of electrical equipment. Today, industrial plants use a lot of switching devices such as inverters and switches of devices that control dynamic mechanisms in

industrial machines [32,35]. The factory floor uses a lot of high-intensity discharge (HID) bulbs to light up the factory. The power source generates harmonics from the above-mentioned devices and the harmonics themselves reduce the performance of those devices. Eliminating harmonics, or minimizing harmonics generated in the power supply, requires new research to improve electrical equipment the response level of which does not create harmonics in the power supply; this is a difficult requirement for researchers. Creating methods to eliminate harmonics in power supplies by computer programs combined with high-tech equipment is also a promising research direction.

In the era of the 4.0 industrial revolution, many nonlinear loads are produced and operated in the distribution power system. Nonlinear loads such as LEDs, computer monitors, power supply switches, and transformers perform the communication between the power source and the loads. These devices generate harmonics in the power supply, harmonics causing significant damage to the performance and operability of the loads [37,38]. Harmonic currents arise in the power supply and seriously affect the communication system [41]. At magnetic couplings in telephones or information transmission sources, harmonics will cause interference and the information transmitted will not meet the requirements or the transmission speed will be delayed [39]. The method that communication equipment suppliers use to minimize harmonics affecting communication lines consists in using equipment to shield the amount of inductance in parallel conductors and building a device to measure and confirm the information interference system. The maximum value of the harmonic current can be much higher than the sine wave shape at the fundamental frequency, causing false tripping [40].

Automation devices use a lot of motors, and the performance of the motors is severely affected by the harmonics generated by the current. Many types of motors operate according to the mechanism of using the PWM method to adjust the operating mechanism; harmonics cause the mechanism to operate not as desired, e.g., torque ripples created by wave interaction harmonics cause this mechanical oscillation [35]. The harmonics generated by the PWM inverters affect the efficiency of the electric motors much more than the power supply [34]. Nonlinear loads in the distribution power supply create levels that negatively affect the performance of transformers. The transformer feeds the rectifier six pulses with a DC load and power dissipation factors such as total harmonic distortion (THD) compromise efficiency in the transformer. Squirrel-bed synchronous motors operate on the flux density at the clearances to increase the torque properties of the motor [35,39]. However, the harmonics generated at those gaps affect the magnetic field of the stator and the rotor negatively, thus impacting the torque of the motor. Researchers calculate the flux density at the gaps using the Finite Element Analysis (FEA) formula. Usually, parallel capacitors are used to perform the function of filtering high-order harmonics or a single-tuned harmonic filter. High-frequency voltage components cause eddy current losses in the core of the AC motor. These losses increase the operating temperature of the fault as well as the coil around the core and can cause undesired torque spikes. Excessive harmonic distortion will cause a lot of zero interference of the current waveform, affecting the timing of the voltage regulator. This may cause the generator to stop working.

## 3. Harmonic Signal Analysis

Harmonic component extraction analysis of the signal is performed in four stages (Figure 3). Stage 1 performs normalization of the signals in the frequency domain or in the time domain. Signal normalization includes many different functions, depending on the type of sensor, so there is no single device that can provide complete normalization for all sensors. Time-frequency representation (TFR) describes parameters performed over time including the instantaneous RMS current parameter, instantaneous fundamental RMS current parameter, total harmonic distortion (THD) parameter, and parameter of instantaneous TnHD. The characteristics of TFR are temporally informative and spectrally informative (Equations (2)–(6)). The signal is analyzed according to the frequency shown through the spectral shapes; the time-varying frequency is shown specifically according to

the time-varying spectral information shape. Time-frequency representation is considered a useful tool for monitoring the signal being analyzed by frequency. Stage 2 performs the estimation of basic signal components and parameters. Instantaneous root-mean-square (RMS) voltages and Instantaneous root-mean-square (RMS) fundamental voltages are the square roots of the mean over one cycle of the square of the instantaneous voltage (Equations (7) and (8)). A quantitative unit is used to measure harmonic distortion in a signal source. Harmonic distortion or total harmonic distortion (THD) is measured as the ratio value of the total power of all harmonic components to the power of the fundamental frequency. The lower the THD value, the more complete the system's output signal waveform is in the sine wave shape and the less noise or distortion there is. The THD value index is used as an indicator of power quality assessment according to the IEEE 519:2014 standard. The smaller the THD value, the less heat generation the power system has, and the lower the thermal power emissions in the field. This proves that the power source is of good quality and improves the performance of electrical equipment. The monitoring and evaluation of power quality can be carried out according to the IEEE 519:2014 standard. Stage 3 performs the classification of signal characteristics. The instantaneous total harmonic distortion (THD) parameter of the harmonics is calculated according to the measure of the harmonic content in a waveform and express value according to Formula (9) and the instantaneous total non-harmonic distortion TnHD(t) parameter of the harmonics is calculated according to Formula (10). In addition, Stage 4 harmonizes the classification of signal types. A deterministic classification method used in practical applications is a rule-based classifier that is easy to implement and relies on threshold settings and expert rules. The classification of harmonic signals is based on parameters for efficient input threshold settings and expert rules that meet IEEE 519:2014 (Figure 4). The harmonic signal in the power supply is normalized to an intensity signal that is analyzed in the frequency and time domains [42,43]. The basic parameters in the harmonic extraction analysis system include RMS fundamental voltage, total waveform distortion, instantaneous RMS voltage, total non-harmonic distortion, and calculated total harmonic distortion [38,39]. The above-stated parameters are used as input parameters for the harmonic component classification and analysis system.

The harmonic signal model is analyzed to extract the signal of the fundamental components of the harmonics according to the IEEE 519:2014 standard [36–39], which is built according to Formula (1) according to the exponential signal complex shape.

$$x_{wd}(t) = e^{j2\pi f_0 t} + A \cdot e^{j2\pi f_1 t} \tag{1}$$

where $f_0$ is the fundamental signal frequency, $f_1$ is the harmonic or interharmonic frequency and t is the time. As for harmonics, $A = 0.25$ and $f_1 = 250$ Hz. As for interharmonics, $A = 0.25$ and $f_1 = 275$ Hz.

Signal time-frequency distribution is a method of representing a signal in terms of time frequencies that include components such as Spectrum, Gabor Transform, and S-Transform [42,43].

The spectrum chart implements the distribution of the fundamental components of the signal in terms of frequency and time. The Hanning window performs a narrow analysis of signal components by the frequency with a window length of 512, and the frequency and time resolution of the signal are performed according to Formula (2).

$$P_x(t, f) = \left| \int_{-\infty}^{\infty} x(t)w(\tau - t) \cdot e^{-j2\pi f t} dt \right|^2 \tag{2}$$

where $x(t)$: signal, $w(n)$: the presence of white noise, $f$: a function of the frequency.

The Gabor Transform method performs the analysis of the local properties of a set of signals with frequency and time domain characteristics [43]. The resolution of the signals

in the frequency domain and in the time domain, always having the same Gabor Transform value for all frequencies, is shown using Formula (3):

$$C(n,k) = \int_{-\infty}^{\infty} x(\tau)h^*(n,k)d\tau \tag{3}$$

where $x(t)$: signal, $h^*(n,k)$: a dual basis of biorthogonal basis.

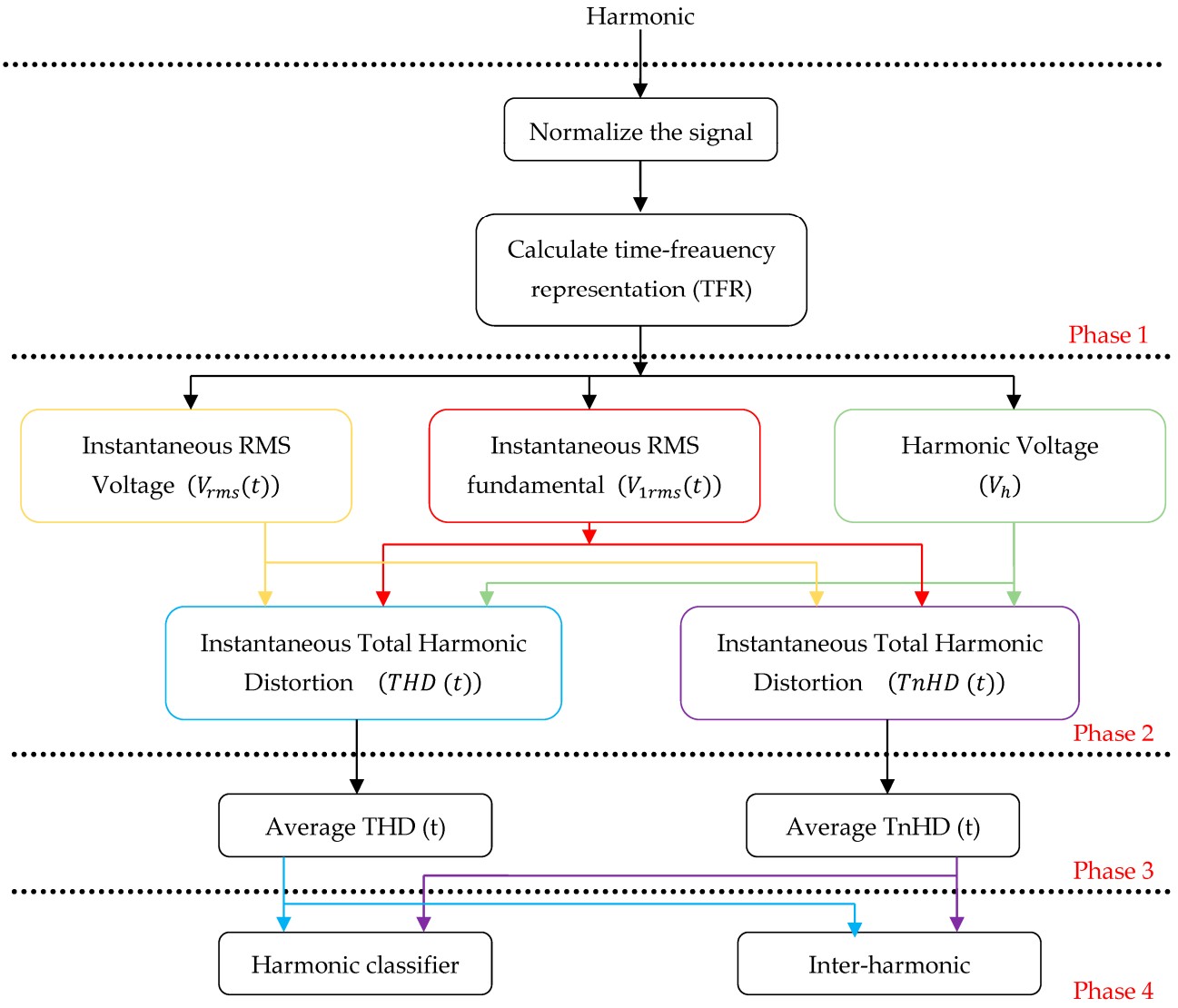

**Figure 3.** Flow chart of harmonic signal detection and classification.

The S-Transform (ST) method is considered as a time-frequency spectral localization method that is made by combining two methods, namely Short-Time Fourier Transform (STFT) and Wavelet Transform. The ST method also uses the window model, but ST implements the method of expanding the windows in Gaussian form and perfecting the signal resolution in the frequency domain represented by the real distribution spectra and virtual shows detailed according to Formulas (4)–(6).

$$ST(\tau,f) = \int_{-\infty}^{\infty} h(t)\frac{|f|}{\sqrt{2\pi}}e^{\frac{-(\tau-t)^2 f^2}{2}} \cdot e^{-j2\pi ft}dt \tag{4}$$

$$g(t) = \frac{1}{\sigma\sqrt{2\pi}} \cdot e^{\frac{-t^2}{2\sigma^2}} \tag{5}$$

$$\sigma(f) = \frac{1}{|f|} \tag{6}$$

where $h(t)$ is the signal, $g(t)$ is the scalable Gaussian window, and $\sigma(f)$ is a control parameter for the Gaussian window.

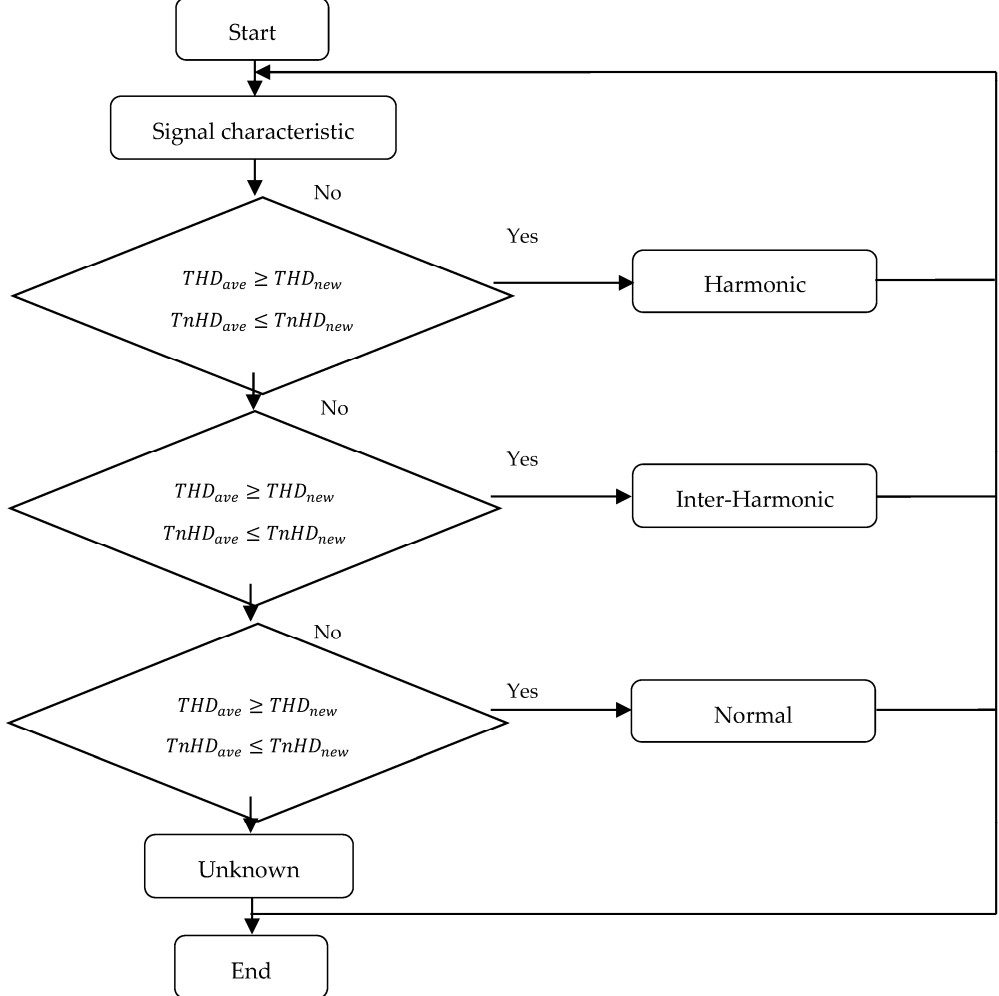

**Figure 4.** The rule-based classifier flow chart for harmonic signals.

The ST method responds well to signals with resolution in the low-frequency domain, and for high-frequency resolution signals the ST method responds well in the time domain. The ST method performs signal component extraction analysis in two types of resolution according to the frequency of different signals. For signals with a high-frequency resolution, the ST method reduces the length of the window, and for signals with a low-frequency resolution the ST method expands the length of the window.

The Instantaneous RMS voltage parameter of the harmonics is calculated according to the root-mean-square (RMS) voltage expressed using Formula (7):

$$V_{rms}(t) = \sqrt{\int_0^{f_x} P_x(t, f) df} \tag{7}$$

where $P_x(t, f)$ is the time-frequency representation signal and $f_s$ is the sampling frequency.

The instantaneous total waveform distortion (TWD) parameter of the harmonics is calculated according to the total waveform distortion relative signal energy existing at a non-fundamental frequency and expressed according to Formula (8):

$$TWD(t) = \frac{\sqrt{V_{rms}(t)^2 - V_{1rms}(t)^2}}{V_{1rms}(t)} \tag{8}$$

where:

$$V_{1rms}(t) = \sqrt{2 \int_{f_{lo}}^{f_{hi}} P_x(t,f)df}, \; f_{hi} = f_0 + 25\text{Hz} \; and \; f_{l0} = f_0 - 25\text{Hz}$$

where $V_{1rms}(t)$ is the instantaneous RMS fundamental voltage; $f_0$ is the fundamental frequency.

The instantaneous total harmonic distortion (THD) parameter of the harmonics is calculated according to the measure of the harmonic content in a waveform and express value according to Formula (9):

$$THD(t) = \frac{\sqrt{\sum_{h=2}^{H} V_{h,rms}(t)^2}}{V_{1rms}(t)} \tag{9}$$

The instantaneous total non-harmonic distortion $TnHD(t)$ parameter of the harmonics is calculated according to the following Formula (10):

$$TnHD(t) = \frac{\sqrt{V_{rms}(t)^2 - \sum_{h=0}^{H} V_{h,rms}(t)^2}}{V_{1rms}(t)} \tag{10}$$

The characteristics of harmonic signals are classified according to their parameters and expressed in detail by the following formulas:

$$V_{rms,ave} = \frac{1}{T} \int_0^T V_{rms}(t)dt$$

Average of total harmonic distortion ($THD_{ave}$):

$$THD_{ave} = \frac{1}{T} \cdot \int_0^T THD(t)dt$$

Total nonharmonic distortion ($TnHD_{ave}$):

$$TnHD_{ave} = \frac{1}{T} \cdot \int_0^T TnHD(t)dt$$

The classification of harmonic signals is based on parameters for efficient input threshold settings and expert rules that meet IEEE 519:2014 (Figure 4).

It measures the performance of analytical methods that extract harmonics in the time domain and the frequency domain by evaluating the feasibility and performance of the methods based on accuracy.

The analysis accuracy is expressed through the accuracy of the signal characteristic measurements, and the mean absolute percentage error (MAPE) value is used as a value to evaluate the accuracy of the physical measurement expressed using Formula (11). The lower the MAPE for the value, the better the performance of the signal characteristic that responds to the measurement.

$$MAPE = \frac{1}{N} \sum_{n=1}^{N} \left| \frac{x_i(n) - x_m(n)}{x_i(n)} \right| \times 100\% \tag{11}$$

where $x_i(n)$ is an actual value, $x_m(n)$ is the measured value, and $N$ is the data number.

Perform analysis on over 100 unique signals in a defined time-frequency domain for accuracy, model complexity, and memory capacity size to select the most efficient harmonic analysis method. Table 2 for the results of the precision analysis indicates that the S-Transform has the highest accuracy. The frequency resolution method for low frequencies is the most suitable.

**Table 2.** The MAPE of the accuracy of the analysis.

| Signal Characteristics | Time-Frequency Domain | | |
|---|---|---|---|
| | **Spectrogram** | **Gabor Transform** | **S-Transform** |
| $V_{rms_{ave}}$ | 0.1571 | 0.627 | 0.0621 |
| $THD_{ave}$ | 0.1541 | 0.963 | 0.0592 |
| $TnHD_{ave}$ | 0.1573 | 0.930 | 0.0593 |

Table 3 shows the results of the analysis of the computational complexity of the model; the results show that the Spectrogram, Gabor Transform, and S-Transform properties are nearly the same. This confirms the choice of the Harmonic Analysis method by windows of the same length and signal for the time-frequency domain.

**Table 3.** The MAPE of the computational complexity of the analysis.

| Signal | Time-Frequency Domain | | |
|---|---|---|---|
| | **Spectrogram** | **Gabor Transform** | **S-Transform** |
| Normal | 20,509,504 | 1,061,408,000 | 21,876,460 |
| Harmonic | 20,504,605 | 1,061,408,000 | 21,876,460 |
| Interharmonic | 20,513,609 | 1,061,408,000 | 21,876,460 |

Table 4 shows the results of the analysis of the memory size. The results show that Gabor Transform gives the smallest memory size, and Spectrogram and S-Transform give the largest memory size.

**Table 4.** The MAPE of the memory size of data analysis.

| Signal | Time-Frequency Domain (Mbyte) | | |
|---|---|---|---|
| | **Spectrogram** | **Gabor Transform** | **S-Transform** |
| Normal | 2,285,356 | 2,240,000 | 2,287,825 |
| Harmonic | 2,285,356 | 2,240,000 | 2,287,825 |
| Interharmonic | 2,285,356 | 2,240,000 | 2,287,825 |

## 4. Harmonic Feature Extraction Technique

The methods of extracting harmonic components in different signal sources in the frequency domain, time domain, and space domain are shown in Figure 5. The time domain harmonic extraction methods include Empirical Mode Decomposition (EMD), which performs the decomposition of the signals into IMF but still ensures the common settings, meeting the goal of separating the high-frequency waves from the X(t) signal and creating symmetrical differential oscillations. Sliding Window EMD (LWEMD) implements $X(t)$ signal separation in operation to separate harmonics from the carrier and shorten the amount of signal filtering by applying Hermite interpolation to detect inflection points at the intersections; the score is 0. Adaptive Harmonic Decomposition (AHD) detects error pulses with a high noise ratio in the time domain and separates error spectra. The AHD performs frequency shifting of the harmonic components, and the adaptive model-based scheme

with a short sliding analysis window (AMS) implements features of online sampling and corrects analytical models directly in the data to extract the harmonic component in real time and monitor the frequency offset at each cycle of the data. The Frequency Domain Harmonic Component Extraction method is the Adaptive Harmonic Wavelet Transform (AHWT), which uses a time-frequency separation technique to exploit harmonic features, cross-compare wavelet features, and identify important features in the signal to determine the efficiency and damage of the structure. Sliding Discrete Wavelet Transform (SDWT) is a frequency domain current control algorithm and performs RC half-cycle correction that halves the delay value and response time; the Spatial Domain Harmonic Component Extraction method is Head-Related Transfer Functions (HRTF), which is a sparse representation of the spherical wavelet basis modeled, consisting of scaling functions at the lowest scale level and wavelet functions at higher levels.

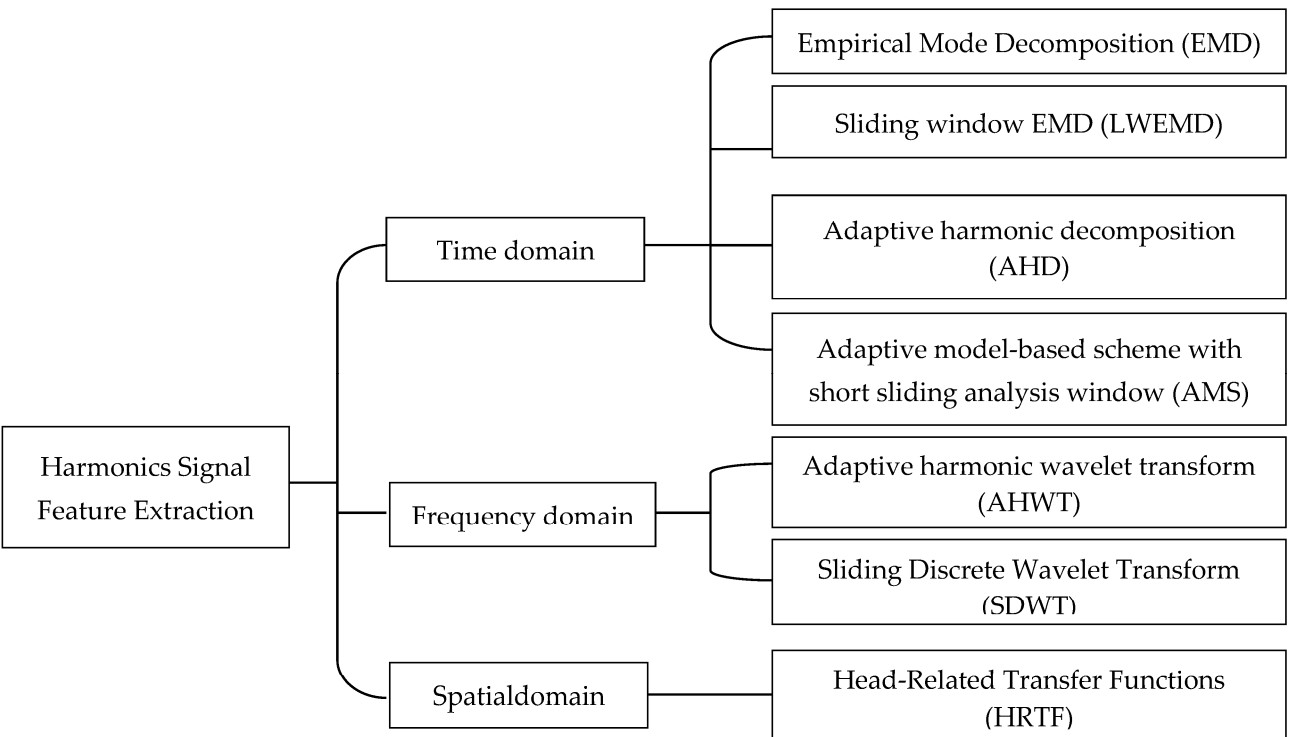

**Figure 5.** Harmonics Signal Feature Extraction techniques.

*4.1. Harmonic Feature Extraction Technique in the Time Domain*

4.1.1. Empirical Mode Decomposition (EMD) Method

This monitors modern systems in real time using sensors that measure unstable signals. The Empirical Mode Decomposition (EMD) method works by intrinsic mode function (IMF) analyzing the multi-component signal in the time domain. Traditional EMD algorithms consider a sample part $x(t)$ of the signal $X(t)$ extracted in terms of $T^-$ *and* $T^+$ according to the two largest and smallest extremes.

A part of signal $x(t)$ oscillates at the starting point, which is the maximum or the minimum, and it passes through the maximum or minimum point and ends with the maximum or minimum, respectively. This is manifested as a high-frequency wave that varies $imf(t)$ at part of the signal $x(t)$. The wave has a low frequency $r(t)$ at part of the signal $x(t)$. The signal $X(t)$ is represented using Formula (12):

$$X(t) = imf(t) + r(t), \ t \in \left(T^-, T^+\right) \tag{12}$$

where $r(t)$ is a residual and can be thought of as some slowly changing behavior.

At the IMF, the response levels of $imf(t)$ high-frequency waves include:

- From a zero-crossing point, the minimum and maximum number of points must be equal or different.
- The amplitude between the maximum and minimum points must be symmetrical to meet the requirement that the mean value of the signal must be 0.

The EMD method performs the analysis of the IMF proficient signals but still ensures the common settings that meet the objective of separating the high-frequency waves from the $X(t)$ signal and symmetrical oscillations. The high-frequency $imf(t)$ is the first IMF level of the $X(t)$ signal. It represents the detected high-frequency wave by the corresponding points of the maximum and minimum point levels. An upper envelope connects the maximum points and a lower envelope connects the minimum points. Spline interpolation between those values is called the extreme point. The sum of the upper and lower bounds is a fixed number. However, the mean value of the signal in the IMF analysis is a constant and is obtained based on Equation (13).

$$r_1(t) = X(t) - imf_1(t) \tag{13}$$

The EMD algorithm is explained in detail in the following steps (Table 5) and the flow chart of EMD is shown in Figure 6.

**Table 5.** The EMD algorithm.

| Step No. | Explained in Detail in the Following Steps |
|---|---|
| Step 1: | All extremes in signal $X(t)$ are detected |
| Step 2: | Connecting all maxima points by spline interpolation to form a contour on $X_{max}(t)$ |
| Step 3: | Connecting all the minimum points by spline interpolation to form the lower contour $X_{min}(t)$ |
| Step 4: | Calculating the average value between the upper and lower contours according to Formula (14) <br> $m(t) = \frac{X_{max}\ (t) - X_{min}\ (t)}{2}$ (14) |
| Step 5: | Finding the value of the high-frequency wave starting at IMF of signal $X(t)$ according to Formula (15) <br> $imf_1(t) = X(t) - m(t)$ (15) |
| Step 6: | The first RF value $imf_1(t)$ is considered the input value of the next screening process. The values of the envelopes and the mean of the high-frequency waves are first deduced. This value is calculated according to Formula (16). This screening process is repeated until the IMF attributes have values that are met <br> $imf_1(t) := imf_1(t) - m(t)$ (16) |
| Step 7: | Reducing the initial signal $X(t)$ according to the mode with the first high-frequency response level and determining the first uniform residual $r_1(t)$ according to the Formula (17) <br> $r_1(t) = X(t) - IMF_1(t)$ (17) |
| Step 8: | The residual value of the first signal part is considered input to the second IMF. This process is repeated until the IMF value is extracted based on Formula (18) <br> $r_i(t) = r_{i-1}(t) - imf_i(t)$ (18) |

The algorithm will terminate when the remainder $r_i(t)$ no longer contains the extreme value. That is, the $X(t)$ signal no longer extracts any more IMF values. In conclusion, the EMD method performs the analysis of $X(t)$ signals into components containing frequency characteristics in descending order through a filtering process. The original signal is analyzed for response according to Formula (19):

$$X(t) = r_n(t) + \sum_{i=1}^{n} imf_i(t) \tag{19}$$

where $n$ is the number of modes.

The EMD method extracts the signal according to the response time domain to solve the frequency resolution problem of the $X(t)$ signal and the EMD algorithm stops when the extreme value is no longer detected. However, the traditional EMD method performs

eight steps of signal filtering to perform full harmonic extraction; the applications at the windows in the signal extraction process are time-consuming and this is the weak point of this EMD method. This results in some unextracted low-frequency signals. The EMD method is not suitable for SAPF filters.

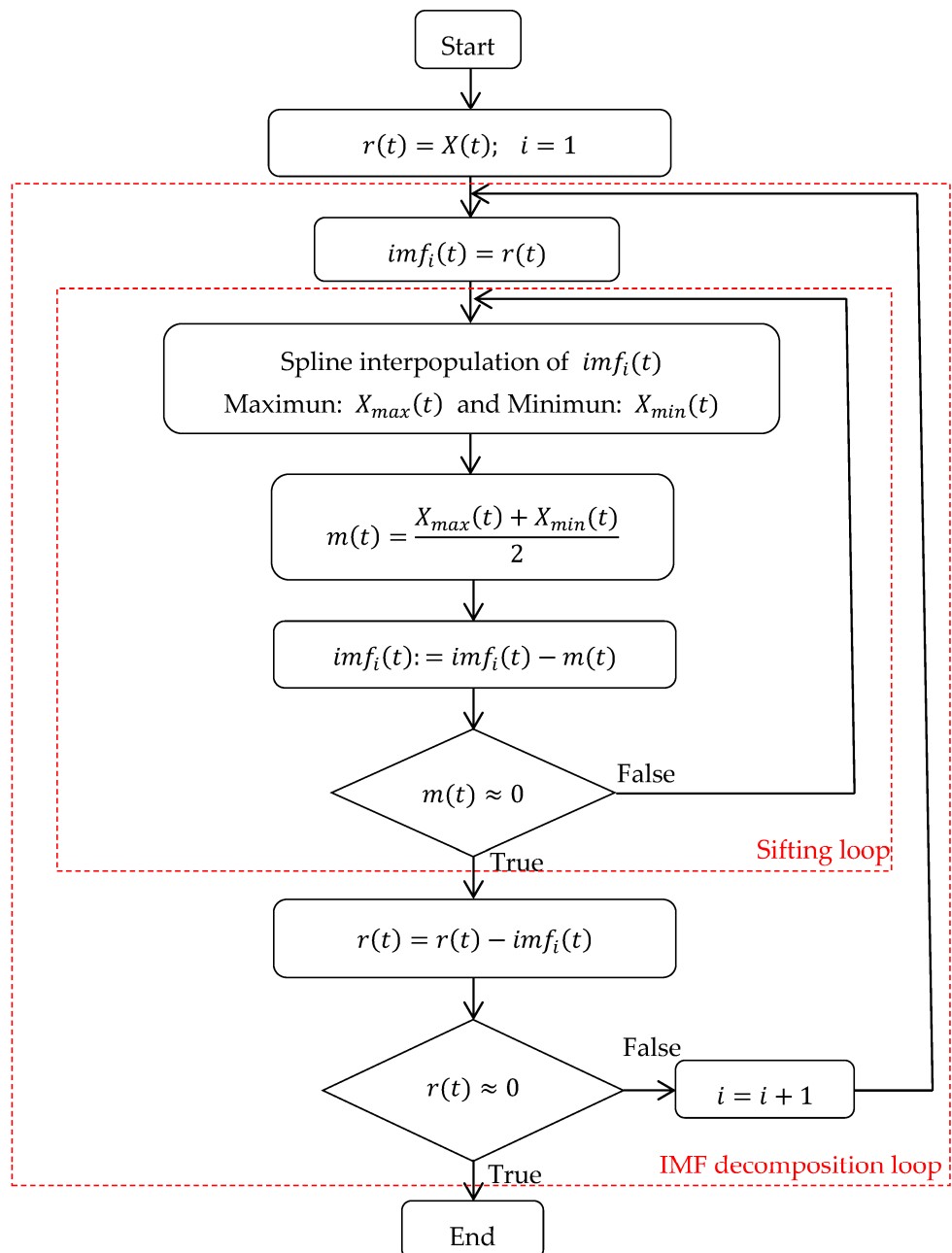

**Figure 6.** Flow chart of EMD Algorithm.

### 4.1.2. Sliding Window EMD (LWEMD)

The LWEMD technique uses a counter that accumulates enough data for signal analysis. The signal screening procedure is performed similarly to the signal screening procedure of the EMD method. The signal analysis data are buffered during the iteration process, securing the signal in the data block and performing the first data block selection analysis to perform a smooth merger. However, the LWEMD method also encounters some challenges when performing signal extraction analysis:

- After each difference iteration, the IMF provides different values at the data blocks. It is necessary to ensure that the IMF value is continuously connected to the data blocks by selecting the number of iterations together.
- The bulk data performed by the screening process make it difficult for real-time analysis of the final signals. Therefore, discarding the final signal is necessary.
- There is a selectively fixed number of iterations for signal filtering. However, low-frequency signals are still present inside the data blocks. Therefore, this low-frequency signal rejection solution should be studied and implemented when removing the harmonic signal from the carrier in the $X(t)$ signal.

The LWEMD method is implemented into $X(t)$ signal analysis to separate harmonics from the carrier. The LWEMD method improves the filtering process from the EMD method by shortening the number of signal iterations by applying the Hermite interpolation to generate an inflection point signal at zero intersections. From here, the envelopes are calculated and all low-frequency signals are cut off for the duration of the algorithm. Some advantages discovered when implementing the LWEMD algorithm in harmonic extraction are as follows:

- The filtering process is streamlined and reduced when implementing the algorithm.
- The detection of low-frequency harmonics in data blocks is guaranteed.
- The execution time to extract harmonics from the signal is less than the traditional EMD method.

The LWEMD algorithm is detailed step-by-step (Table 6) and the flow chart applying LWEMD to harmonic extraction is shown in Figure 7.

**Table 6.** The LWEMD algorithm.

| Step No. | Explained in Detail Step-by-Step |
|---|---|
| Step 1: | Creating data block from buffer 1 dataset. |
| Step 2: | Implementing the Hermite spline interpolation method for the buffer region of the signal $X(t)$. |
| Step 3: | Calculating the average value according to Formula (20). $$m_1(t) = \frac{X_{max}(t) - X_{min}(t)}{2} \qquad (20)$$ |
| Step 4: | Finding the IMF value at the first data block of the signal according to Formula (21). $$imf_i(t) = X(t) - m_1(t) \qquad (21)$$ |
| Step 5: | Calculating the residual value of the signal in the first signal data block according to Formula (22). $$r_1(t) = X(t) - imf_1(t) \qquad (22)$$ |
| Step 6: | Obtaining the value of $r_1(t)$ |
| Step 7: | Applying interpolation Hermite spline to detect extremes and calculating contours according to Formula (23). $$dr_1(t) = \frac{dr_1(t)}{dt} \qquad (23)$$ |
| Step 8: | Extracting the value at time i where no $dr_1(t)$ value exists. |
| Step 9: | Calculating the mean value of signal $m_2(t)$ according to the residual value of signal $r_1(t)$ according to Formula (24). $$m_2(t) = \frac{r_{1max}(t) - r_{1min}(t)}{2} \qquad (24)$$ |
| Step 10: | Finding the IMF value at signal value $m_2(t)$ according to Formula (25). $$imf_2(t) = r_1(t) - m_2(t) \qquad (25)$$ |
| Step 11: | Calculating the residual value at signal value $m_2(t)$ according to Formula (26). $$r_2(t) = r_1(t) - imf_2(t) \qquad (26)$$ |
| Step 12: | Cutting off the block signal at time i. |
| Step 13: | The data are saved in the last block and at least seven extreme points are saved for duplicate data blocks. |

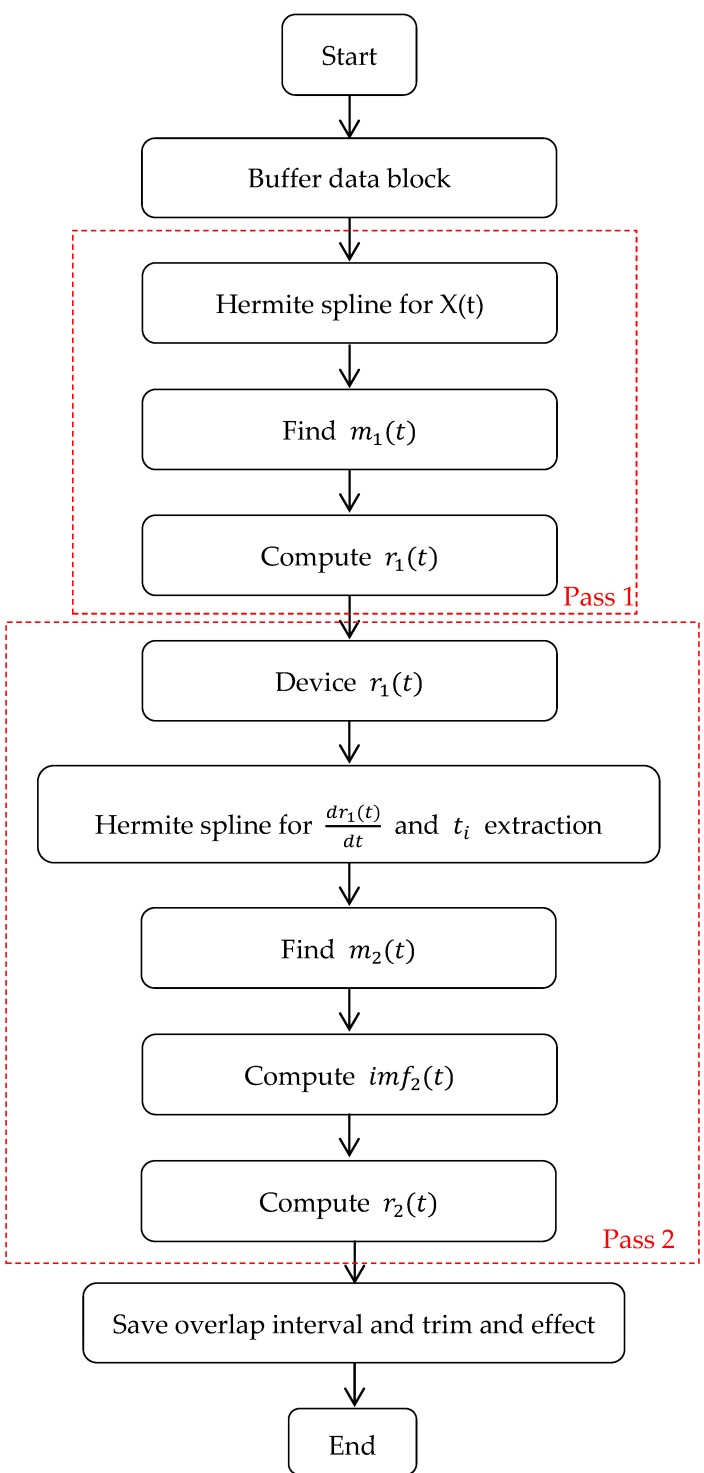

**Figure 7.** Flow chart of LWEMD for harmonic extraction.

### 4.1.3. Adaptive Harmonic Decomposition (AHD) Method

Vibration signals from rotating motors contain error pulses that give rise to frequency modulation effects. In the frequency domain at a uniform interval, there is a connection between two pulses including the fault-causing pulse and the pulse of a harmonic cluster. In a definite resonance sequence, this symphony helps determine the placement of samples and spectrum distribution of the signal and detects periodic pulses that cause errors. The adaptive harmonic decomposition method detects error pulses with a high noise ratio in

the time domain and discrete error spectra even in the case of low signal-to-noise (SNR). The error pulse detection algorithm is detailed in the following steps (Table 7).

**Table 7.** The error pulse detection algorithm.

| Step No. | Explained in Detail in the Following Steps | |
|---|---|---|
| Step 1: | The camcorder generates vibration pulses that form error pulses and is mathematically modeled for the error pulse according to Formula (27). $$g(t) = s(t) + n(t) = A(t).cos\left[2\pi \int_0^t f(\tau)d\tau + \varphi_0\right] + n(t) =$$ $$= \sum_{p=1}^{p} S_p(t) + n(t) = \sum_{p=1}^{p} a_p.cos(2\pi f_p(t) + \theta_p) + n(t)$$ Where $s(t)$ is an impulse signal consisting of harmonics. P has a frequency in the resonant series g(t) with IA and IF modulation levels. $n(t)$ represents additional noise signals and noise components arising from the source of the camera vibrations. | (27) |
| Step 2: | An error pulse detection model is designed by separating the $S_p(t)$ harmonics in the resonant range of the $g(t)$ angular signal and putting them back together. This proposed harmonic extraction method can extract harmonics with an SNR cap that responds to component decomposition and bandwidth shrinkage. | |
| Step 3: | The adaptive harmonic decomposition method performs frequency shifting of the harmonic components. Given an initial frequency $f_p$ of the series resonant to the Pth harmonic part, the $S_p(t)$ harmonic of the moment with the Pth harmonic component is transformed according to Formula (28). $$S_p(t) = U_p(t).cos(2\pi f_p(t)) + V_p(t).sin(2\pi f_p(t))$$ (28) The two harmonic displacement components are described using Formula (29) $$\begin{cases} U_p(t) = a_p.cos\left[2\pi\left(f_p - \tilde{f}_p\right)(t) + \theta_p\right] \\ V_p(t) = -a_p.sin\left[2\pi\left(f_p - \tilde{f}_p\right)(t) + \theta_p\right] \end{cases}$$ (29) | |
| Step 4: | where the estimated frequency $\tilde{f}_p$ is close to the original frequency $f_p$ value. The two shifting harmonics $U_p(t)$, $V_p(t)$ form good variation patterns in the time domain and noise to zero-frequency trends in the frequency domain. The $S_p(t)$ harmonic, at the time of having the Pth harmonic component, is reconstructed to the original amplitude and phase according to Formulas (30) and (31). $$a_{(p|\tilde{f}_p=f_p)} = \sqrt{U_p^2(t) - V_p^2(t)}$$ (30) $$\theta_{(p|\tilde{f}_p=f_p)} = tan^{-1}\left[\frac{-V_p(t)}{U_p(t)}\right]$$ (31) | |
| Step 5: | Based on the aforementioned frequency change operation, optimization is performed followed by discretization according to the estimated frequency value and the $S_p(t)$ harmonic component is reproduced in the Pth harmonic according to Formula (32): $$\begin{array}{c} min \\ \{U_p\},\{V_p\},\{\tilde{f}_p\} \end{array}\left\{\tau_\vartheta\left(U_p, V_p, \tilde{f}_p\right)\right\} =$$ (32) $$\begin{array}{c} min \\ \{U_p\},\{V_p\},\{\tilde{f}_p\} \end{array}\left\{\varnothing\omega_{P_2}^2 + \varnothing V_{P_2}^2 + \vartheta g - (C_p U_p + S_p.V_p)_2^2\right\}$$ where $\varnothing$ second order deviation operator is used to calculate a quantitative value for the smoothness of harmonic displacement components $U_p(t)$, $V_p(t)$. $\vartheta$ Penalty coefficient and discrete variables g sampled over time $\{t_0, t_1, \ldots, t_i, \ldots, t_{l-1}\}$ are calculated according to Formulas (33)–(36). $$g = [g(t_0), g(t_1), \ldots, g(t_{l-1})]^T$$ (33) $$U_P = [U_p(t_0), U_p(t_1), \ldots, U_p(t_{l-1})]^T$$ (34) $$V_P = [V_p(t_0), V_p(t_1), \ldots, V_p(t_{l-1})]^T$$ (35) $$C_p = diag\left[cos\left(2\pi\tilde{f}_p(t_0)\right), cos\left(2\pi\tilde{f}_p(t_1)\right), \ldots, cos\left(2\pi\tilde{f}_p(t_{l-1})\right)\right]$$ (36) $$S_p = diag\left[sin\left(2\pi\tilde{f}_p(t_0)\right), sin\left(2\pi\tilde{f}_p(t_1)\right), \ldots, sin\left(2\pi\tilde{f}_p(t_{l-1})\right)\right]$$ (37) Analyzing the $S_p(t)$ harmonic component at the time of the Pth harmonic by updating, and updating the details of the optimal equation according to Formula (37). | |

4.1.4. Adaptive Model-Based Scheme with Short Sliding Analysis Window (AMS)

This improves power quality by providing the correct and sufficient amount of current loss compensation in the power supply. It fully and in detail determines the fundamental and harmonic components of a power supply that condition the efficiency of the power

supply's lossy current compensation operation. The weakness of the frequency domain method of harmonic extraction is that it generates a sampling delay of at least one cycle and depends on the frequency resolution. The weakness of the method of extracting harmonic components in the power source by the time domain method is that it does not guarantee the stability and deviation of the interference.

The proposed adaptive model-based scheme with a short sliding analysis window method fulfills the features of online sampling and directly corrects analysis models in the data to extract the harmonic component in real time and monitor the frequency offset at each cycle of the sample data. It extracts the harmonic component accurately, providing timely compensation for the loss of current, improving the power quality, and improving the working efficiency of the new method. It extracts the fundamental and harmonic components of sample data online. It sets the point of the harmonic signal to follow the sine wave shape.

The power signal ($S$) in discrete time ($S_n$) form of the amount of sample collected (N) during the ($\Delta t$) period is presented as a sine component (H) according to Formula (38):

$$S_n = \sum_{h=1}^{H} a_h.cos(nhw_1\Delta t + \theta_h), \ n = 0, \ 1, \ \ldots, \ N-1 \tag{38}$$

where $a_h$: amplitude, $\theta_h$: initial phase angle, $w_1 = 2\pi f_1$: fundamental angular frequency.

To simplify the calculation, Formula (1) is analyzed according to Formula (39):

$$S_n = \sum_{h=1}^{H} \left( A_h.e^{jnhw_1\Delta t} + A_h^*.e^{-jnhw_1\Delta t} \right) = \sum_{h=1}^{H} \left( A_h.x_h^n + A_h^*.(x_h^n)^* \right) \tag{39}$$

where $A_h = \frac{a_h e^{jnhw_1\Delta t}}{2}$: complex amplitude, $x_h = e^{jnhw_1\Delta t}$, and $(*)$: complex conjugate calculation.

The amplitude value ($A$) calculated by minimizing the error between the actual number of samples, $s_n$, and its estimate is presented using Formula (40):

$$A = \mathrm{arg}min\left( \sum_{n=0}^{N-1} |S_n - \hat{S}_n|^2 \right) \tag{40}$$

Complex amplitude estimation is according to Formula (41):

$$\hat{A} = \left( X^T X \right)^{-1}.X^T.S \tag{41}$$

where $T$: Transpose of a matrix.

The amplitude and phase angle of the h-th harmonic are arguments of the complex amplitude and are expressed by Equations (42) and (43):

$$a_h = 2|A_h| \tag{42}$$

$$\theta_h = arg\{A_h\} \tag{43}$$

Harmonic component extraction generates a minimal error because the fundamental frequencies of the signals in the power supply are time-biased and deviated from their nominal values due to the power imbalance between the power supply and the loads on demand. There are many methods used to modify the X matrix when frequency bias occurs. The Frequency Domain Interpolation (FDI) method analyzes and detects the fundamental frequency in a better manner. However, frequency resolution and delay of at least one cycle are incurred depending on the finite amount of the analysis window in FDI. The Kalman filtering method and the PLL-based technique perform estimation error tracking and time-domain parameter tuning of the system to perform the synchronization of the results from

the measurement. However, the weakness of this method is that the method of determining the parameters is not suitable for maintaining the stability of the signal quantity and improving the accuracy and convergence speed. It prevents long durations arising in frequency domain methods and creates numerical instability in time domain methods. The short sliding window technique performs signal analysis ($s_n$) from Equation (2) to form a low-pass filter according to Equation (44), described as follows:

$$s_{1-n} = A_1 x_1^n + A_1^*.(x_1^n)^* \tag{44}$$

observing three consecutive data samples and describing them in detail according to Equation (45):

$$
\begin{aligned}
s_{1-n-2} &= A_1.x_1^{n-2} + A_1^*.\left(x_1^{n-2}\right)^* \\
s_{1-n-1} &= A_1.x_1^{n-1} + A_1^*.\left(x_1^{n-1}\right)^* \\
s_{1-n} &= A_1 x_1^n + A_1^*.\left(x_1^n\right)^*
\end{aligned}
\tag{45}
$$

The assumed linear relationship of three consecutive samples is shown in Equation (46). The fitting parameter ($\varepsilon$) is considered the error estimation parameter described by Equation (47), with the estimated sample being $\hat{s}_{1-n}$.

$$\hat{s}_{1-n} = s_{1-n-2} + \varepsilon.s_{1-n-1} \tag{46}$$

$$
\begin{aligned}
\varepsilon &= \mathrm{arg}min(E) = \mathrm{arg}min\left( \sum_{n=3}^{N} |s_{1-n} - \hat{s}_{1-n}|^2 \right) = \\
&= \mathrm{arg}min\left( \sum_{n=3}^{N} |s_{1-n} - s_{1-n-2} - \varepsilon.s_{1-n-1}|^2 \right)
\end{aligned}
\tag{47}
$$

The minimum error estimate ($E$) is presented by Equation (48) and then the linear estimation parameter is reconstructed according to Equation (49):

$$\frac{dE}{d\varepsilon} = 2 \sum_{n=3}^{N} (s_{1-n} - s_{1-n-2} - \varepsilon.s_{1-n-1}).(-s_{1-n-1}) = 0 \tag{48}$$

$$\varepsilon = \frac{\sum_{n=3}^{N}(s_{1-n-1}).(s_{1-n} - s_{1-n-2})}{\sum_{n=3}^{N}(s_{1-n-1}^2)} \tag{49}$$

Representative equations of three samples are reconstructed according to Equation (50):

$$
\begin{aligned}
A_1.x_1^{n-2} + A_1^*.\left(x_1^{n-2}\right)^* + \varepsilon.A_1.x_1^{n-1} + \varepsilon.A_1^*.\left(x_1^{n-1}\right)^* &= \\
= A_1.x_1^n + A_1^*.(x_1^n)^* => x_1^2 - \varepsilon.x_1 - 1 &= 0
\end{aligned}
\tag{50}
$$

The fundamental frequency information of the three sample signals is shown at $x_1$ according to Equation (51):

$$x_1 = \frac{\varepsilon \pm j\sqrt{\varepsilon^2 + 4}}{2} = e^{jw_1\Delta t} = cos(w_1\Delta t) + jsin(w_1\Delta t) \tag{51}$$

The fundamental frequency components of the three sample signals containing the matching parameter ($\epsilon$) are shown by Equation (52):

$$f_1 = \frac{cos^{-1}\left( \frac{\sum_{n=3}^{N}(s_{1-n-1})(s_{1-n} - s_{1-n-2})}{2\sum_{n=3}^{N} s_{1-n-1}^2} \right)}{2\pi\Delta t} \tag{52}$$

Fundamental frequency value ($f_1$) performs an analysis model tuning operation conducting the X-matrix modification function, which improves the accuracy of the fundamental frequency and harmonic component extraction in the signal.

The benefits of the AMS solution are that it monitors frequencies by sliding window function and performs analysis model modification to improve the accuracy of real-time varying fundamental frequency and harmonic extraction. The sliding window in N online acquisition samples helps in normal frequency domain signal analysis and frequency monitoring. The harmonic extraction is performed quickly, regardless of frequency resolution. The fundamental frequency component ($f_1$) prevents the numerical imbalance of conventional time domain techniques and helps to determine the appropriate parameters (Figure 8).

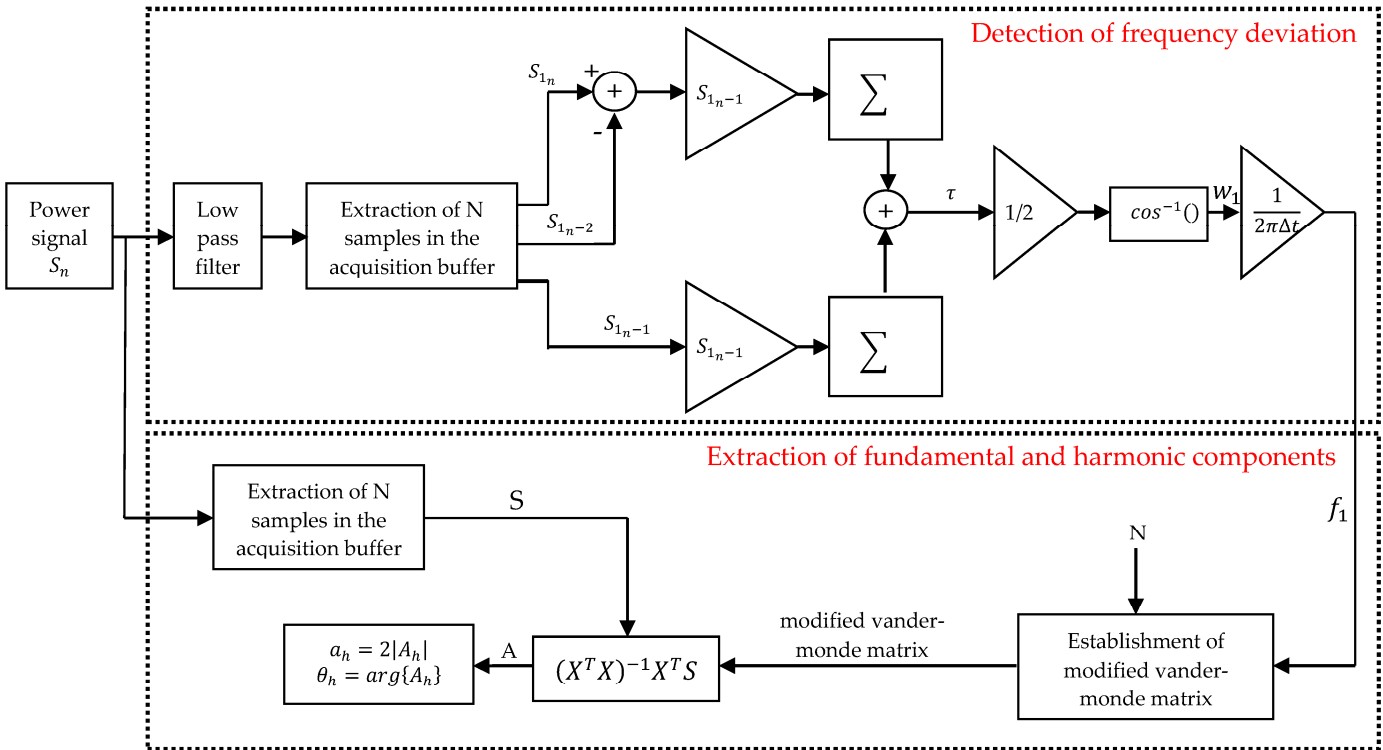

**Figure 8.** Solution procedure of the proposed AMS.

The limitation of the AMS method is that it uses a low-pass filter that defines the parameters to match the frequency response to the fundamental signal due to the attenuation of the signal magnitude. The study of modifying frequency detection by a new method that is better than the low-pass filter method is promising.

### 4.2. Harmonic Feature Extraction Technique in the Frequency Domain

#### 4.2.1. Adaptive Harmonic Wavelet Transform (AHWT)

The components of a wave signal include frequency content or time frequency. Adaptive harmonic wavelet transform uses a time-frequency separation technique to exploit highly efficient response features and outperforms empirical mode decomposition (EMD) methods. The AHWT method uses a deterministic basis to extract the features of the signal in the time-frequency domain. The cross-compared wavelet and AHWT characteristics confirm important features in wave signals to determine the efficiency and damage of the waveform structure. The AHWT method implements one filter bank; at each filter, a specific frequency range ($m2\pi$, $n2\pi$), $0 \leq m \leq n$ is designed, which is called the parameter level. The size of each filter is designed to be small and complete in the frequency domain, also

known as the ideal sequence pass filter. A complete filter forms an orthogonal wavelet and is detailed by Equation (53):

$$w_{mnk}(t) = w_{mn}\left(t - \frac{k}{n-m}\right) = \frac{exp\left[in2\pi\left(t - \frac{k}{n-m}\right)\right] - exp\left[im2\pi\left(t - \frac{k}{n-m}\right)\right]}{(n-m)i2\pi(t)} \quad (53)$$

The value of the generalized harmonic wavelet is obtained by the inverse Fourier transform according to Formula (54):

$$w_{mnk}(w) = \begin{cases} \frac{1}{(n-m)2\pi}e^{-1w\frac{k}{n-m}} & , m2\pi \le w \le n2\pi \\ 0 & , otherwise \end{cases} \quad (54)$$

where the integer $K$ is the displacement parameter in the region $(m, n)$ and each level of the $K$ value represents a frequency range in the frequency domain. The advantage of the harmonic wavelet method is that the signal is analyzed within a limited range of specific frequency ranges.

The discrete harmonic wavelet transform is based on the Fast Fourier Transform (FFT) method, which responds well to the signals of sensors operating in real-world environments that collect time series signal data $\{x(r), r = 0, 1, 2, \ldots, N-1\}$; Fourier coefficients $\{F(q), q = 0, 1, \ldots, N-1\}$ and $F(q)$ are calculated using the Fast Fourier Transform (FFT) Formula (55).

$$F(q) = \frac{1}{N}\sum_{r=0}^{N-1} x(r).exp\left(-\frac{i2\pi rq}{N}\right) \quad (55)$$

The harmonic wavelet coefficient $\{a_{mnk}\}$ is calculated using Formula (56).

$$a_{mnk} = \sum_{l=0}^{n-m-1} x(r).exp\left(-\frac{i2\pi kl}{n-m}\right), \ k = 0, 1, \ldots, n-m-1 \quad (56)$$

This study reconstructs the original time series from the parameters for the harmonic wavelets function. However, in discrete transform, continuous wavelet functions are replaced by corresponding circular continuous functions according to Formula (57):

$$W_{mnk}^C(r) = \frac{1}{(n-m)}\sum_{l=m}^{n-1} exp\left(i2\pi l\left(\frac{r}{n} - \frac{k}{n-m}\right)\right) \quad (57)$$

The signal $S(r)$ is determined in the time unit interval according to Formula (58):

$$S(r) = \sum_{k=m}^{n-1}\left\{a_{mnk}.W_{mnk}^C(v) + \overline{a}_{mnk}.\overline{W}_{mnk}^C(r)\right\} \quad (58)$$

The selection of $\{(m_0, n_0), (m_1, n_1), \ldots, (m_{l-1}, n_{l-1})\}$ parameter pairs must begin with the $m_0 = 0$ value and continue with each pair that touches each other until $n_{l-1} = N_f$. $N_f$ is the Nyquist frequency and $l$ is the total number of levels.

The strength of the harmonic wavelet lies in the flexible selection of parameter pairs (m, n) as the basis for the possible subharmonics. In a case where a wavelet level (m, n) is determined in a frequency band, that signal is separated by the Wavelet Transform method. This demonstrates that the Harmonic Wavelet Transform method has the potential to perform the same detection. The point of this issue is what method to use to choose the parameter pair (m, n) accordingly. According to the signal processing theory, a signal whose signal energy is sparsely concentrated in a few basic functions is considered a good signal. The method of Shannon entropy according to Formula (59) is implemented by the original Harmonic Wavelet Transform Hybrid Improvement method to select a suitable pair of (m,

n) parameters. Each pair of $\{(m_0, n_0), (m_1, n_1), \ldots, (m_{l-1}, n_{l-1})\}$ parameters selected for processing in the algorithm is considered an $\varnothing = \{0, 1, \ldots, N-1\}$ element and searches for the best region that meets the wavelet coefficient with the minimum entropy value:

$$H(Z) = -\sum_j P_j . log P_j \tag{59}$$

where $P_j = \frac{|Z_j|^2}{\|Z\|^2}$ *and* $P_j . log P_j = 0$ *when* $P_j = 0$.

The Shannon entropy value is a measure of sparsity value and the smaller the Shannon entropy value, the better the search area. The search loop occurs typically two or three times. Eight Fourier coefficients are used in the loop, and the number of iterations for 16 elements is used for the algorithm of mathematical equations at level 2 (Figure 9).

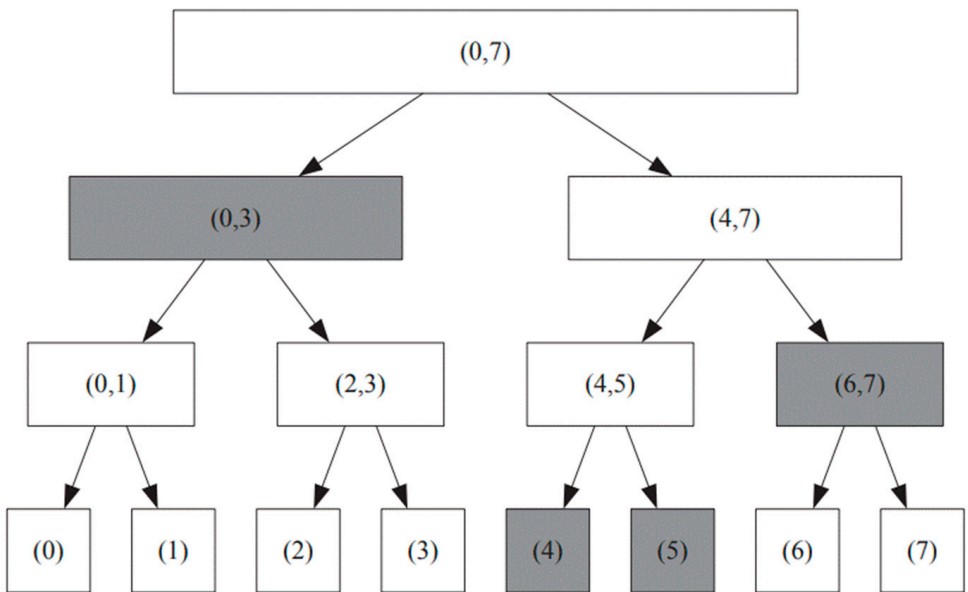

**Figure 9.** A sample binary search tree for partition selection used for AHWT (N = 16).

In the algorithm flowchart (Figure 10), each Fourier coefficient is set for the first group, where in the Shannon entropy value is set as the first entropy value. Each parameter pair (*m*, *n*) represents a subgroup. In the second iteration, the FFT is performed on each group of two adjacent Fourier coefficients, and the values of the entropy are calculated and compared with the sum of the corresponding initial entropy. Through the whole process of searching by multiple loops, the pairing process price is the best. The AHWT coefficient is updated. At the same time, the wavelet's basic function reimplements the signal reconstruction.

### 4.2.2. Sliding Discrete Wavelet Transform (SDWT)

The active power filter (APF) has a conversion frequency from 10 kHz to 20 kHz. At the output of the APF, an LCL filter is used for a good response to the group of converted harmonics. However, the LCL filter has complex design parameters, often generating resonance points, and the complex circuit design and control algorithm for the APF becomes difficult. Designing SiC-MOSFET into the source device of the APF to increase the switching frequency to 50 kHz and using an L filter instead of the LCL filter helps suppress subharmonics at the switching switch to a minimum, making the circuit design simple and the algorithm in the APF easy and simple. At the same time, the sampling frequency and switching frequency are faster and increase exponentially. As a result, harmonic detection achieves higher accuracy and better output current control. The harmonic extraction algorithm applied to the APF is represented by a sliding window discrete Fourier transform (SDFT) and frequency domain analyzed flow control algorithm. The modification of the

SDWT algorithm half cycle and RC half cycle reduces algorithm delay by half and APF dynamic response time from two times to half of the power frequency cycle.

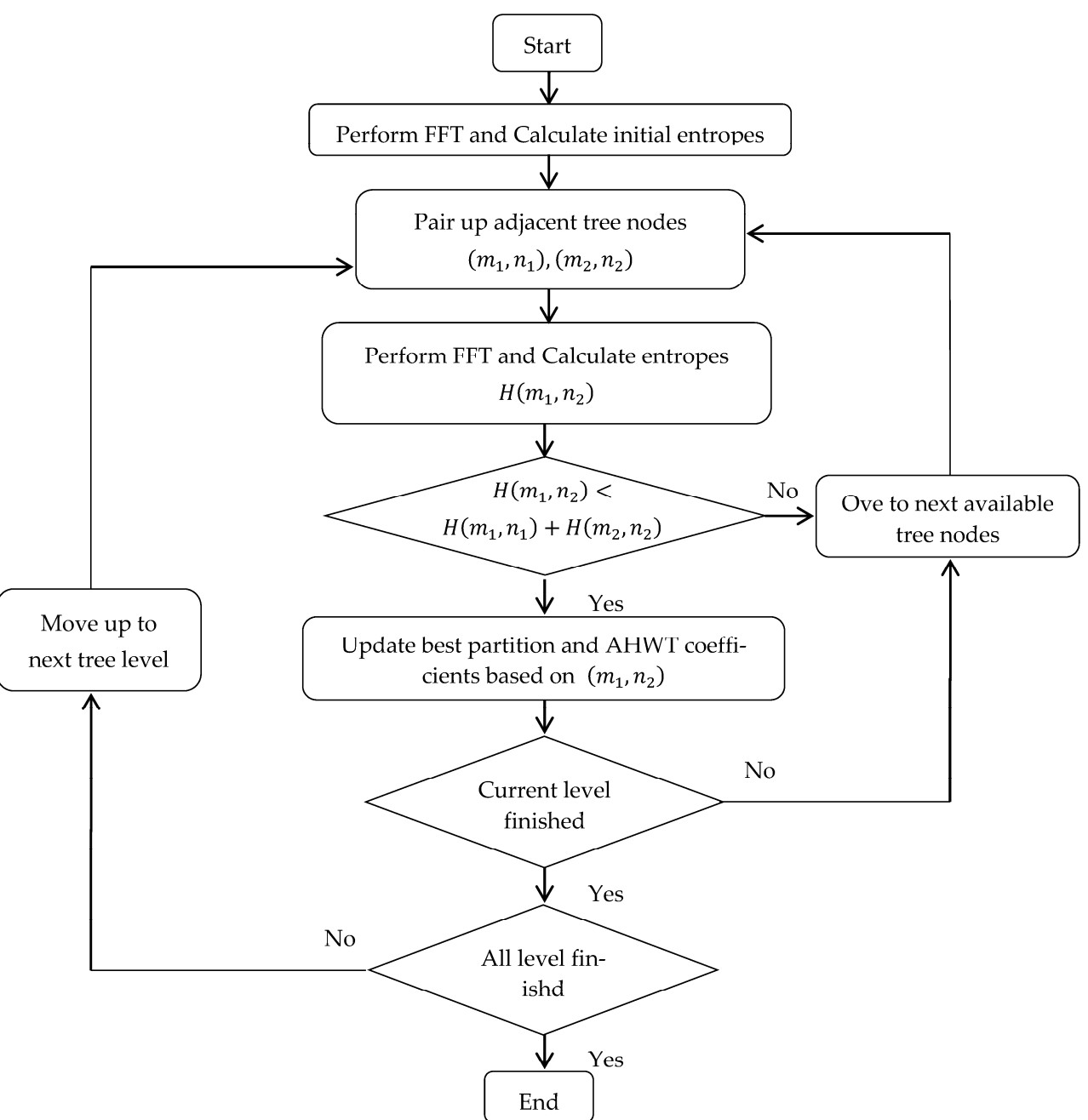

**Figure 10.** Flow chart of the iterative searching algorithm.

The harmonic extraction algorithm plays a decisive role in the harmonic mitigation model. Precise harmonic extraction helps the APF to provide accurate and fast compensating currents. The SDWT method is a commonly used harmonic extraction and is developed from the traditional DWT method. The formulas used for the harmonic extraction process are shown below.

1.  The description of the nth harmonic component is made according to Formula (60):

$$i_n(k) = A_n(k).cos\left(\frac{2\pi nk}{N}\right) + B_n(k).sin\left(\frac{2\pi k}{N}\right) \tag{60}$$

2. The AA and BB coefficients are calculated according to Formula (61):

$$\begin{cases} A_n(k) = A_n(k-1) + \frac{2}{N}.cos\left(\frac{2\pi nk}{N}\right)[i(k) - i(k-N)] \\ B_n(k) = B_n(k-1) + \frac{2}{N}sin\left(\frac{2\pi nk}{N}\right)[i(k) - i(k-N)] \end{cases} \tag{61}$$

where $N$ is the number of sampling points in one cycle and $K$ is the latest current sampling point.

The main difference between the SDWT method and the DWT method is in the different update method of the coefficients $A_n(k)$, $B_n(k)$. The DWT method (Figure 11) requires a full data sampling cycle to calculate and update the coefficients. The SDWT method (Figure 12) takes a new sample each time and the corresponding sample value of one cycle before it is discarded and replaced with the newly acquired sample value and updated coefficients, which increases the time efficiency of the system.

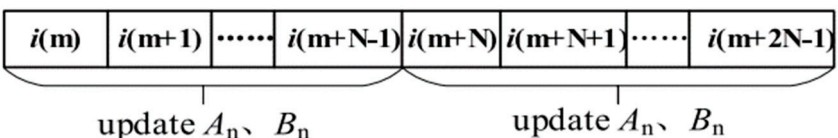

**Figure 11.** Data sampling cycle of the DWT method.

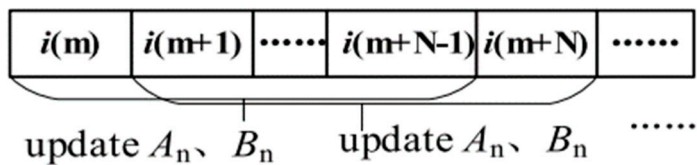

**Figure 12.** Data sampling cycle of the SDWT method.

However, SDWT requires additional memory space to store sample values for one cycle, and this introduces its inherent delay over one cycle.

The extraction of half-cycle harmonics by the SDWT method is described by an exponential function and nth harmonic expression according to Formula (62):

$$i_n(k) = I_n(k)e^{j\frac{2\pi k}{N}} \tag{62}$$

The $I_n(k)$ coefficient is calculated according to Formula (63):

$$I_n(k) = \frac{1}{N}\sum_{l=k-N+1}^{k} i(l)e^{-j\frac{2\pi nk}{N}} =$$
$$= I_n(k-1) + \frac{1}{N}i(k)e^{-j\frac{2\pi nk}{N}} - \frac{1}{N}i(k-N)e^{-j\frac{2\pi nk}{N}}.e^{j2n\pi} \tag{63}$$

If you multiply both sides of Equation (5) by the expression $e^{j\frac{2\pi k}{N}}$, you obtain the result of Equation (64):

$$i_n(k) = i_n(k-1)e^{j\frac{2\pi k}{N}} + \frac{1}{N}i(k) - \frac{1}{N}i(k-N)e^{j2n\pi} \tag{64}$$

If you convert Formula (6) to the Z domain, you obtain the SDFT transfer function according to Formula (65):

$$H_s(Z) = Z\left(\frac{i_n(k)}{i(k)}\right) = \frac{1}{N}.\frac{1 - Z^{-N}}{1 - e^{j\frac{2n\pi}{N}}.Z^{-1}} \tag{65}$$

Formula (7) is broken down into the expressions in Equation (66):

$$H_s(Z) = \frac{1}{N} H_c(Z).H_{rn}(Z); \quad H_c(Z) = 1 - z^{-N}; \quad H_{rn}(Z) = \frac{1}{1 - e^{j\frac{2n\pi}{N}}.Z^{-1}} \tag{66}$$

where $H_c(Z)$ is divided into a filter by frequency and the results are expressed according to the Bode plot (Figure 13) and the $H_c(Z)$ formula is further subdivided into Formula (67):

$$H_c(Z) = 1 - z^{-N} = \prod_{K=0}^{N-1} \left( 1 - e^{j\frac{2n\pi}{N}}.Z^{-1} \right) \tag{67}$$

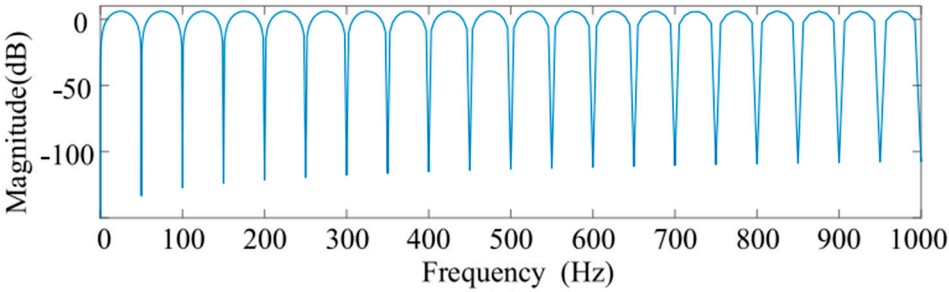

**Figure 13.** Bode diagram of traditional SDFT.

Formula (9) shows that the filter has N 0 points and each of these 0 points represents one attenuation for the system where harmonics are generated, or it causes a sampling delay. This proves that a signal with N zeros means that there are N sampling delays, also known as delayed sampling periods.

$H_{rn}(Z)$ is considered a resonator and increases with the nth harmonic level. In addition to filters and resonators, other necessary frequency components can be extracted. The main harmonic components generated in the power supply are odd harmonic components. In cases where the harmonic component is ignored, Formula (68) is rewritten as follows:

$$H_c(Z) = \prod_{K=0}^{\frac{N}{2}-1} 1 - e^{j\frac{2n\pi(2k+1)}{N}}.z^{-1} = 1 - e^{jn\pi}.Z^{-\frac{N}{2}} = 1 + Z^{-\frac{N}{2}} \tag{68}$$

The filter according to Equation (10) contains only N/2, which demonstrates only the attenuation of the odd harmonic component, and the delay is reduced from one period to one half cycle (Figure 14).

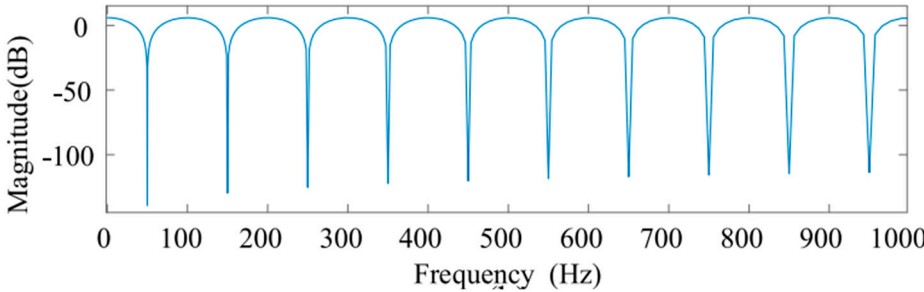

**Figure 14.** Bode diagram of a half-cycle SDFT.

Within the bandwidth limitation, the PI algorithm cannot separate and monitor harmonics well. The periodic repetition of the harmonic currents based on the internal principle provides better harmonic monitoring. Link delay in the repeater controller has an impact on the dynamic response performance of the controller. By deploying the SDFT method in

process optimization, the controller repeats the half cycle to reduce the sampling delay by half a cycle (Figure 15).

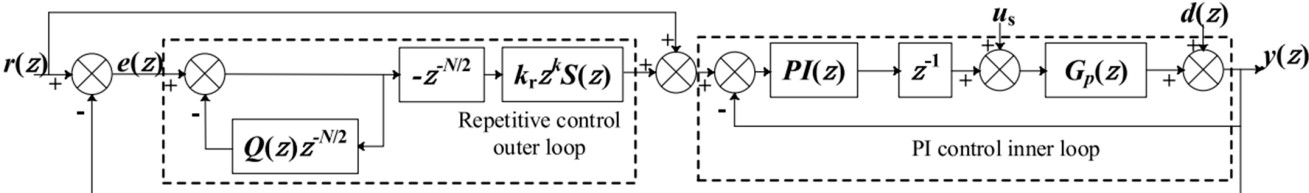

**Figure 15.** Structure of PI + repetitive controller.

Here, $H_s$: feedback signal from voltage network, $d(z)$: internal noise signal, $G_p(z)$: object to be controlled, $K_r$: scale factor, $z^k$: phase compensation link, $S(z)$: low pass filter.

The harmonic current in the THD power supply network is 4.15, which meets the IEEE 519:2014 standard. However, the half-cycle control algorithm makes the second harmonic high. This requires new research to improve the performance of shunt adaptive power filters.

### 4.2.3. Head-Related Transfer Function (HRTF) Methods

The head-related transfer function (HRTF) methods are built in the spatial domain based on the structure of spherical wavelets. The HRTF method represents local features with a small number of analytic functions that allow the spatial resolution to be controlled in the local region in the sphere with control coefficients. The HRTF spatial transformation models perform the harmonic decomposition of spheres that represent the rough structure and respond to the lowest level of the model. The HRTF method is formed from a combination of spherical harmonics and spherical wavelets that respond to the corresponding coarse structure and spatial detail. The HRTF method performs the function of describing the audio transmission characteristics of the spatial audio signal source. The HRTF method is a variable of many parameters such as frequency, direction, and distance, and the HRTF implements the principle that spherical wavelets are local functions and wavelets have difficulty discretely expanding the signals in spheres.

The objective function of the spatial domain HRTF modeling is built according to the objective function $H(\theta, \varnothing)$ determining the magnitude in the direction $(\theta, \varnothing)$ with azimuth angle $\theta \in (-180^0, 180^0)$ and elevation angle $\varnothing \in (-90^0, 90^0)$. The incidences $(0^0, 0^0)$ *and* $(90^0, 0^0)$ represent the front and left directions. A point source $\vec{r} = (r, \theta, \varnothing)$ with its distance r in the coordinate system (Figure 16).

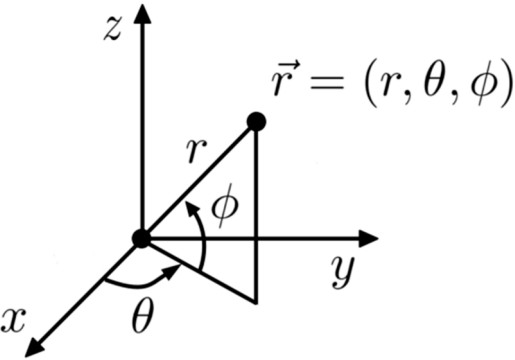

**Figure 16.** Spherical coordinate system.

The objective function $H(\theta, \varphi)$ of the HRTF is built as weighted sums of the spherical values that implement the real values of the harmonics shown using Formula (69):

$$H(\theta, \varnothing) = \sum_{n,m} Y_n^m(\theta, \varnothing).hs_n^m \tag{69}$$

where $Y_n^m(\theta, \varnothing)$ is a harmonic function of a sphere of order n and of mode m calculated according to Formula (70). $hs_n^m$ is the corresponding coefficient implemented according to the overall description of the spatial frequency of the objective function.

$$Y_n^m(\theta, \varnothing) = \begin{cases} (-1)^{m+1}\sqrt{\frac{2n+1}{2\pi} \cdot \frac{(n+m)!}{(n-m)!}}.P_n^{-m}.sin(\varnothing).cos(m\theta) \ m > 0 \\ \sqrt{\frac{2n+1}{4\pi}}.P_n^0.sin(\theta) \ m = 0 \\ (-1)^m\sqrt{\frac{2n+1}{2\pi} \cdot \frac{(n-m)!}{(n+m)!}}.P_n^m.sin(\varnothing).cos(m\theta) \ m < 0 \end{cases} \tag{70}$$

where $P_n^m$ is the associated Legendre function of order n and mode m.

The harmonic coefficient $(h_n^m)$ for the continuous objective function $H(\theta, \varnothing)$ on the sphere is calculated using Formula (71):

$$h_n^m = \int_{-\pi}^{\pi} \int_{-\frac{\pi}{2}}^{\frac{\pi}{2}} H(\theta, \varnothing).Y_n^m(\theta, \varnothing)sin(\varnothing)d\varnothing d\theta \tag{71}$$

The vector (H) of the HRTF dataset is calculated using Formula (72) based on the matrix of spherical harmonics (*W*) and the extended vector coefficients (*C*):

$$H = W.C + \epsilon \tag{72}$$

The objective function of the sphere in the spherical wavelet functions is based on the functional weights calculated using Formula (73):

$$H(\theta, \varnothing) = \sum_k \partial_{0,k}.\varphi_{0,k}(\theta, \varnothing) + \sum_{j \geq 0} \sum_i \gamma_{j,i}.\varphi_{j,i}(\theta, \varnothing) \tag{73}$$

where $\partial_{0,k}$: approximation coefficients, $\gamma_{j,i}$: wavelet or detail coefficients.

$$\partial_{j,k} = \sum_{l \in K(j)} \widetilde{h}_{j,k,l}\partial_j + 1,l$$

$$\gamma_{j,i} = \sum_{l \in I(j)} \widetilde{g}_{j,i,l}\partial_j + 1,l$$

where $\widetilde{h} \ and \ \widetilde{g}$: decomposition filters.

The approximation coefficient uses Formula (74):

$$\partial_{j+1,k} = \sum_{k \epsilon K(j)} h_{j,k,l}\partial_{j,l} + \sum_{i \in I(i)} g_{j,i,l}\gamma_{j,i} \tag{74}$$

The HRTF method performs the transformation and decomposition of the objective function down to the lowest level value of 0 to meet high accuracy in a certain locality and performs data size compression to ensure that the data size is maintained. Several coefficients expand the corresponding coefficients.

The HRTF method performs a holistic description of the harmonics according to the spatial features in all directions of the HRTF, and the spherical wavelet model performs the capture of the local features of the harmonic signal. The HRTF method performs spherical harmonic analysis and kie3m controls the spatial resolution to underestimate perceived importance. Spherical harmonics represent the coarse structure of low resolution, and

spherical wavelets perform the modeling of spatial details. The rough structure $H_c(\theta, \varnothing)$ implemented in the objective function of HRTF is calculated according to Formula (75):

$$H_c(\theta, \varnothing) = \sum_{n=0}^{N_c} \sum_{|m| \leq n} Sh_n^m(\theta, \varnothing).hc_n^m \tag{75}$$

where $hc_n^m$: expansion coefficient of spherical harmonic, $Sh_n^m(\theta, \varnothing)$: spherical harmonic.

The residual part $H_d(\theta, \varnothing)$ is the difference between the initial HRTF intensities $H_c(\theta, \varnothing)$ and the rough structure $H_c(\theta, \varnothing)$ calculated using Formula (76):

$$H_d(\theta, \varnothing) = \sum_{l}^{L_d} \sum_{i}^{I_l} Sw_l^i(\theta, \varnothing).hr_l^i \tag{76}$$

where $I_l$: truncated level scale, $hr_l^i$: corresponding coefficient, $Sw_l^i(\theta, \varnothing)$: spherical wavelet.

The approximated coarse structure of the HRTF method is carried out according to Formula (77):

$$H_{com}(\theta, \varnothing) = H_c(\theta, \varnothing) + H_d(\theta, \varnothing) \tag{77}$$

The HRTF method performs intensity based on spherical wavelets and shows low-order spherical harmonics with a very small number of parameters that perform the modeling of finer details in the residuals.

## 5. Comparison High Light on Harmonic Feature Extraction Technique and Future Research Topic

This paper reviews the literature on methods of extracting harmonic components in the time domain, frequency domain, and space domain from previous reviews by the authors and methods that have not been previously evaluated. This study tries to conduct a review of the literature, specifically each mathematical formula, and, step by step, perform the harmonic extraction of the methods of extracting harmonic components in the time domain, frequency domain, and spatial domain. The strength of this study is that it describes, in detail and step by step, the harmonic component extraction according to each corresponding mathematical model; it effectively evaluates the signal extraction time and the performance of the signal for each method of extracting harmonic components. The strength of time-domain extraction methods is that, using a window frame to extract signals in small amounts and monitor low frequencies, it helps to improve the signal extraction performance that other signal extraction methods have. Traditional brands are not available. The weakness of time domain extraction methods is that the signal processing time is still slow. As for the improvement of the signal extraction performance in terms of time, the Active Distribution Rejection Control (ADRC) method appears to be promising. The Frequency Domain and Space Domain Signal Extraction method is conducted according to the wavelet formula, which requires a lot of mathematical modeling, and this is a weak point and is difficult for computers to implement, and the users also need to perform very difficult computer programming.

The EMD method uses the intrinsic mode function to process multi-component signals in the time domain [44,45]. The EMD reviews each signal sample according to the maximum and minimum two extremes [46]. This gives rise to variable high-frequency and low-frequency waves in some parts of the signal since the intrinsic mode function only responds to high-frequency signal levels [47,48]. The EMD performs signal filtering in seven steps to complete harmonic component extraction; the applications at windows during signal filtering take a lot of time, and this results in low-frequency harmonic signals that are not fully extracted. The data LWEMD method uses the LWEMD signal analysis response using a cumulative buffer [44,49]. The data used to analyze the signal are buffered during the iteration [47]. However, since the IMF values provide different values after each iteration, an integrated method is needed to remove the last signal of the iterations; the low frequency is still present in the signal after the iteration at different times in the

data block [50,51]. The advantage of the LWEMD method is that the time to perform signal extraction is shortened compared to the EMD method [52]. The LWEMD method is proposed to be used in active distribution rejection control to eliminate harmonics. The AHD method performs the formation of a design error pulse detection model by detecting the harmonics in the resonant array with the original signal by frequency shifting the harmonic components [53–55]. The AHWT method is designed with a bank of signal filters and, at each filter, the frequencies are designed according to a specific filter; the size of each filter is designed to be small according to the frequency domain [56–59]. The AHWT method performs the feature extraction of each signal in the frequency-time domain according to wavelet features [60–62]. The AHWT method performs the selection of parameter pairs to respond well to the wavelet search area according to the minimum entropy value [63,64]. The Difference Evolution (DE) method, which optimizes each parameter pair to provide the AHWT method as a direction to consider for research in this field, is the proposed artificial intelligence method for parameter pair selection [65,66]. The SDWT method performs the half-cycle correction of the signal, which minimizes the delay [62] and reduces the signal response time to half a cycle of the source frequency [67,68]. The SDWT method performs harmonic shifting in the frequency domain [69]. The SDWT method re-updates the sample value after each iteration and discards the previous sample value [70]. However, the SDWT method increases the second harmonic when performing half-cycle frequency control [71–73]. The method to suppress second order harmonics is a promising research direction for the future. The AMS method performs online sampling and corrects the models directly in the data to extract real-time harmonic component and frequency offset per cycle [56,60–62]. The AMS method prevents generation duration and instability in the frequency domain [63]. However, a low-pass filter is used to determine suitable frequency response parameters [64–66]. A new study that needs frequency modification in a low-pass filter is promising for the future. A brief description of the strengths and weaknesses of the methods to extract the harmonic component of the signal is shown in Table 8.

**Table 8.** Comparison of harmonic signal feature extraction processing techniques.

| Method | Advantages | Disadvantages | Reference List |
|---|---|---|---|
| EMD | • Meets the objective of separating the high-frequency waves. | • The extreme value is no longer detected.<br>• The signal extraction process is time-consuming.<br>• Some unextracted low-frequency signals.<br>• The EMD method is not suitable for SAPF filters. | [44–48] |
| LWEMD | • Accumulates enough data for signal analysis.<br>• The filtering process is streamlined and reduced when implementing the algorithm.<br>• Guaranteed detection of low-frequency harmonics in data blocks.<br>• The execution time to extract harmonics from the signal is less than the traditional EMD method. | • Low-frequency signals are still present inside the data blocks.<br>• Iterations after each difference. The IMF gives different values to the data blocks.<br>• The bulk data performed by the screening process make it difficult for real-time analysis of the final signals. | [44,47,49–52] |
| AHD | • Determines the placement of samples and spectrum distribution of the signal and detects periodic pulses that cause errors.<br>• Detects error pulses with a high noise ratio in the time domain and discrete error spectra even in the case of low signal-to-noise (SNR). | • The connection between two pulses including the fault-causing pulse and the pulse of a harmonic cluster. | [53–55,57–59] |

**Table 8.** *Cont.*

| Method | Advantages | Disadvantages | Reference List |
|---|---|---|---|
| AMS | • Fulfills the features of online sampling and corrects analysis models directly in the data to extract the harmonic component in real time and monitor the frequency offset at each cycle of the sample data.<br>• Provides timely compensation for the loss of current, improves the power quality, and improves the working efficiency of the new method. | • Generates a sampling delay of at least one cycle and depends on the frequency resolution.<br>• Not guaranteed for the stability and deviation of the interference.<br>• Uses a low-pass filter that defines the parameters to match the frequency response to the fundamental signal due to the attenuation of the signal magnitude. | [56,60–66] |
| AHWT | • Uses a time-frequency separation technique to exploit highly efficient response features and outperforms empirical mode decomposition (EMD) methods.<br>• Deterministic basis to extract the features of the signal in the time-frequency domain.<br>• Flexible selection of parameter pairs (m, n) as the basis for the possible sub-harmonics. | • The point of this issue is what method to use for choosing the parameter pair (m, n) accordingly.<br>• The original harmonic wavelet transform hybrid improvement method is to select a suitable pair of (m, n) parameters. | [62,67–73] |
| SDWT | • Half-cycle and RC half-cycle reduce algorithm delay by half.<br>• Provides accurate and fast compensating currents. | • The half-cycle control algorithm makes the second harmonic high. | [67,68,74–81] |
| HRTF | • Represents local features with a small number of analytic functions.<br>• Performs the function of describing the audio transmission characteristics of the spatial audio signal source. | • Represents the rough structure and responds to the lowest level of the model.<br>• A variable of many parameters such as frequency, direction, and distance; HRTF implements the principle that spherical wavelets are local functions and wavelets have difficulty discretely expanding the signals in spheres.<br>• Spherical harmonics represent coarse structures of low resolution, and spherical wavelets perform the modeling of spatial details. | [49,82–84] |

The challenges for the methods to perform the extraction of harmonic components in the signal include:

- Choosing the correct and proper harmonic components in the signal. Harmonic parts have low frequency and high frequency, an even harmonic type and an odd order type. At present, no signal extraction method meets the aforementioned criteria. The correct and precise selection of the harmonic component in the signal is still an open issue for future scientific researchers.
- Mathematical formulas using harmonic component extraction in the signal are often very complex, making it difficult for computer programmers as well as causing delays in the processing of computer programs. There have not been many studies applying bio-inspired optimization methods to perform harmonic extraction, and this is an open road for researchers. The mathematical models in bio-inspired optimizations are very simple and convenient for computer programmers.

- The extraction time of the harmonic component in the signal contributes to improving the efficiency of harmonic filtering devices. Current methods do not meet the harmonic filtering rate required by computer processing because there is no perfect method to perform the extraction of all harmonic components in the signal.
- The efficiency of extracting harmonic components in the signal is the deciding factor in the success or failure of the harmonic filter device. Accurate, complete, and timely harmonic component extraction is a suitable input for classifiers and signal selectors that compensate for harmonic losses quickly. A new method of applying artificial intelligence algorithms is considered perfect for future researchers.

## 6. Conclusions

Signals are recognized and classified based on the feature composition of each corresponding signal type. The accuracy of signal feature extraction methods contributes to improved signal-processing performance. The nonlinear signals are feature-extracted based on the nonlinear dynamic analysis method and this nonlinear dynamic analysis method is widely used in signal processing. The selection of signal-processing algorithms and features determines the performance of the signal-processing system. The main contributions of this study are as follows:

- This paper presents four methods (EMD, LWEMD, AHD, and AMS) to extract harmonic components based on the time domain. The AMD method performs full harmonic extraction by binning, but it is time-consuming in terms of signal processing and many low-frequency signals exist. The LWEMD method performs a small amount of signal extraction during filtering, which improves the efficiency of signal extraction and shortens the signal processing time, effectively responding to low-frequency signal monitoring. The AHD method detects pulses with a high noise rate. The AMS method implements the analysis model modification in the sliding window to improve the harmonic extraction accuracy.
- The study also presents, in detail, two extraction methods (AHWT, SDWT) to output harmonics in the frequency domain. The AHWT method uses a cross-comparison of wavelet features to monitor the efficiency and damage of the signal waveform structure. The SDWT method performs half-cycle correction, which reduces the signal processing time but generates high-gain second harmonics.
- This paper presents a method (HRTF) for spatial domain harmonic component extraction. The HRTF states that spherical wavelets are local functions and have difficulty expanding signals in spheres, while spherical harmonics represent coarse structures of low resolution. Both are used to model spatial details.

This paper makes an overview and presents in detail some methods used to extract the harmonic and fundamental wave components in signals in the time domain, frequency domain, and space domain. In principle, all three applications of harmonic and fundamental frequency extraction are different. In some cases of different signals, extracting the harmonic component gives better results. However, the frequency domain extraction method requires a more complex method and uses a lot of complex mathematical equations, which leads to higher computational costs and an increased sampling delay of a minimum of one sampling cycle and depends on the resolution of the frequency. A method is used to extract the harmonic component in the frequency domain and halve the sampling period to shorten the harmonic extraction time. However, it generates second harmonics. Along with the development of switching devices as well as frequency modulation devices, this leads to an increasing amount of digital signal processing over time and the development of new technologies and processing methods. Signal management is increasingly demanding in terms of the characteristics of harmonic extraction efficiency; the signal extraction processing time must be fast. A method of signal extraction in the time domain is used by many developers of harmonic extraction applications because they help in the real-time monitoring of the signal. In the time domain, the signal extraction method is used in combination with methods such as feedback control techniques for

signals, filtering, and canceling techniques according to the user's requirements. However, the feedback signal tuning technique is a difficult technique that requires high accuracy of the tuning parameters; therefore, the signal-processing system needs to perform the correct parameter correction to achieve high signal processing efficiency that provides optimal signal-processing performance when there are changes in signal processing. Extracting the harmonic component signal in the time domain has one weakness: it does not guarantee the stability and durability of the generated noise signals.

Filters and harmonic component signal extraction always come with many challenges that signal extraction or processing techniques face, such as the delay or lengthening of the feedback signal when processing signal filtering; to perform filter cancellation, the delay signal must be within a certain frame of reference. Several methods have been developed to eliminate the aforementioned problems, and those methods are implemented entirely in a stationary frame of reference that ensures the complete dynamics of their operation. Mathematical equations are also considered for use to simulate filtering or canceling processes for deferred signals because the operators of the difference are close to the mathematical model. The operator models used to cancel the reflected signals in this period are implemented at discrete impulse response filters. Techniques have been developed to cancel the signal delay using first-order filters to fourth-order filters and they are widely used in digital signal processing and deliver solid performance. Moreover, error states arise when signal processing is lower or zero; the response rate has the shortest time to meet the goal of extracting the most accurate harmonic component for filter system selection to compensate for the best harmonic losses and improve the performance of the signal system.

Extracting harmonics from the signal by artificial intelligence techniques is increasingly being considered. Using meta-heuristic optimization techniques in signal processing with support from computer science is always a promising research direction.

**Author Contributions:** Conceptualization, M.L.D. and P.B.; methodology, M.L.D.; software, M.L.D.; validation, M.L.D. and P.B.; formal analysis, M.L.D.; investigation, M.L.D.; resources, M.L.D.; data curation, M.L.D.; writing—original draft preparation, M.L.D.; writing—review and editing, M.L.D.; visualization, M.L.D.; supervision, P.B. and R.M.; project administration, P.B. and R.M.; funding acquisition, P.B. and R.M. All authors have read and agreed to the published version of the manuscript.

**Funding:** This work was supported in part by the Ministry of Education of the Czech Republic (Project No. SP2023/090).

**Data Availability Statement:** Not applicable.

**Acknowledgments:** The authors are extremely grateful to VSB–Technical University of Ostrava, Czechia for financial support. They would also like to express their gratitude to Van Lang University, Vietnam, for supporting this research.

**Conflicts of Interest:** The authors declare no conflict of interest.

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
