# Peer review of "Harmonics Signal Feature Extraction Techniques: A Review"

_mathematics, doi:10.3390/math11081877_

Round 1

Reviewer 1 Report

The paper presents review of harmonics signal feature extraction techniques.  Below I attached some questions, editorial mistakes etc., to which the Authors should to assume an attitude:

1. In line 13, there is necessary to add numbers of cited references ?. Please revised it.

2. Moreover, the abstract not indicated in detail what is it in the review paper. The sentence in line 19-21 should be revised. This sentence is not necessary.

4. The quality of the Figures 3,4,8  have to be improved.

5. The Figures 5, 10, seems to be prepared in a rush. It should be improved with utilization more professional software.

6. In whole paper, the dots in the figures and tables captions should be after number.

7. It is necessary to add the references in table 2?

8. The conclusion should be extended to present more in depth the signal feature extraction techniques. Please revied this section.

9. Moreover, the English language should be improved.  I suggest to revised it with the assistance of a native speaker.

Author Response

Dear Reviewer,

First, we would like to express our greatest gratitude to you, the editorial team and the reviewers whose valuable comments made on this paper have significantly improved the quality of this paper. We have addressed all the comments made in the new version of the paper and will list the changes made item by item in response to these comments below.

Thanks, and Best Regards
Minh Ly Duc

Reviewer 2 Report

1. In this study, techniques to extract fundamental frequencies and harmonics in the frequency domain, in the time domain and in the spatial domain include literature review and overall assessment. The authors provide an overview of the fundamental frequency and harmonic extraction methods in recent years, analysis and presentation of their advantages and limitations.

2.     In the figure2, effects of harmonics should be demonstrated in detail.

3.     In the figure3, Flow chart of harmonic signal detection and classification should be demonstrated in detail.

4.     In the figure5, harmonics Signal Feature Extraction techniques should be demonstrated in detail.

5.     Revise the English thoroughly before submission.

Author Response

(The authors gave the same response as above.)

Reviewer 3 Report

Section 1 must be improved.

-       Authors should emphasize contribution and novelty, the introduction needs to clarify the motivation, challenges, contribution, objectives, and significance/implication. 

-       You should introduce the problem in more detail so that the reader is immediately clear about the purpose of your study. Specify better the essential elements of the problem.

-       What are the advantages of the proposed work in comparison to already existing ones? This must be clear in the text. Please compare the proposed work with other existing ones.

-       25) “main components in the power signal” in signal processing?

-       60) “overall harmonic detection method” there only two as you wrote

-       85) “FA” Do not use acronyms until you have presented the full definition

Section 2 can be improved.

-       112) Figure 2 must be improved: The text is too small and appears blurry, making it difficult for the reader to follow the flow of information.

-       Also you should improve the content of the Figure caption. Add information, remember that the reader should retrieve all information for reading the figure right from the caption. So it must be complete and comprehensive.

Section 3 must be improved.

-       You must properly introduce the equation, list in detail the variables contained in it with a concise description of the meaning. To make them more readable show them in a bulleted list. In this way the reader will be able to understand the contribution of each variable.

-       201) “Spectrogram method” Introduce adequately the topic

-       Figure 4 must be improved: The text is too small and appears blurry, making it difficult for the reader to follow the flow of information.

-        259-263) Why you have introduced only MAPE metrics? This section is essential in order to demonstrate the effectiveness of a methodology. Furthermore, only by adopting adequate metrics will it be possible to compare results with those obtained by other researchers.

Section 4 must be improved.

-       287) Figure 5must be improved: Correct the caption, furthermore you use a lot of acronyms without the definitions, do not use acronyms until you have presented the full definition

-       presents the algorithms as a sequence of steps, you can use a table (as table 1) or a numbered list to help you

-       You must properly introduce the equation, list in detail the variables contained in it with a concise description of the meaning. To make them more readable show them in a bulleted list. In this way the reader will be able to understand the contribution of each variable.

-       558) Table 1 is not sufficiently clear, there are many formulas, it’s convenient to introduce the equation early in the text and then recall these equation in the algorithm presentation

-       647) Figure 8 must be improved: The text is too small and appears blurry, making it difficult for the reader to follow the flow of information.

-       754) Make Figure and Caption fit on the same page.

-       755) Correct the caption of all figures, for example replace Figure .10 with Figure 10.  I will not repeat this advice again, it also applies to the other occurrences.

Section 5 must be improved.

-       It is suggested that the authors give the research challenges and hotspots in the review.

-       There are other reviews about it, the author should cite them and explain how his article differs from them and what his contribution is in relation to already published knowledge.

-       The comparison scheme needs to be improved so that it is more readable for the reader.

-       A detailed discussion of the paper reviewed is missing. Try to summarize what was obtained and try to extract useful information from the work carried out. Also add bibliographic references to support your conclusions, to give more weight to your statements.

Author Response

(The authors gave the same response as above.)

Round 2

Reviewer 1 Report

The paper was revised in accordance with the review.

Reviewer 3 Report

The authors addressed the reviewer's comments with attention and modified the paper with the suggestions provided. The new version of the paper has improved both in the presentation and in the contents